**Data Availability Statement:** The EMIS-2017 dataset used for this analysis has been obtained

# Changes in the prevalence of self-reported sexually transmitted bacterial infections from 2010 and 2017 in two large European samples of men having sex with men–is it time to re-evaluate STI-screening as a control strategy?

Ulrich Marcus[1]*, Massimo Mirandola[2], Susanne B. Schink[1], Lorenzo Gios[2], Axel J. Schmidt[3,4]

**1** Department for Infectious Diseases Epidemiology, Robert Koch Institute, Berlin, Germany, **2** Department of Diagnostics and Public Health, Infectious Diseases Section, University of Verona, Verona, Italy, **3** Sigma Research, London School of Hygiene and Tropical Medicine, London, United Kingdom, **4** Division of Infectious Diseases, Cantonal Hospital St. Gallen, St. Gallen, Switzerland

* MarcusU@rki.de

## Abstract

### Background/Objectives

Many European countries reported increased numbers of syphilis, gonorrhoea and chlamydia diagnoses among men who have sex with men (MSM) in recent years. Behaviour changes and increased testing are thought to drive these increases.

### Methods

In 2010 and 2017, two large online surveys for MSM in Europe (EMIS-2010, EMIS-2017) collected self-reported data on STI diagnoses in the previous 12 months, diagnostic procedures, STI symptoms when testing, number of sexual partners, and sexual behaviours such as condom use during the last intercourse with a non-steady partner in 46 European countries. Multivariate regression models were used to analyse factors associated with diagnoses of syphilis, gonorrhoea/chlamydia, and respective diagnoses classified as symptomatic and asymptomatic. If applicable, they included country-level screening rates.

### Results

Questions on STI diagnoses and sexual behaviours were answered by 156,018 (2010) and 125,837 (2017) participants. Between 2010 and 2017, overall diagnoses with gonorrhoea/chlamydia and syphilis increased by 76% and 83% across countries. Increases were more pronounced for asymptomatic compared to symptomatic infections. The proportion of respondents screened and the frequency of screening grew considerably. Condomless anal intercourse with the last non-steady partner rose by 62%; self-reported partner numbers grew. Increased syphilis diagnoses were largely explained by behavioural changes (including more frequent screening). Gonorrhoea/chlamydia increases were mainly explained by

from the London School of Hygiene and Tropical Medicine under a data transfer agreement that prohibits to sharing the dataset publicly. Although we cannot make study data publicly accessible at the time of publication, all authors commit to make the data underlying the findings of the study available in compliance with the PLOS Data Availability Policy. Data requests should be addressed to the London School of Hygiene and Tropical Medicine Research Operations Office Data Management Lead: alex.hollander@lshtm.ac.uk, the first author (MarcusU@rki.de), and the Principal Investigator of EMIS-2017 (Peter. Weatherburn@lshtm.ac.uk). Individuals requesting data should present their research objective(s) and enclose a list of requested variables. To protect the confidentiality of participants, data sharing is contingent upon appropriate data handling and good scientific practice by the person requesting the data and should furthermore be in accordance with all applicable local requirements. The London School of Hygiene and Tropical Medicine administrative offices are located at Keppel Street, London WC1E 7HT, United Kingdom.

**Funding:** UM, service contract 2015 71 01 between Robert Koch-Institute and Consumers, Health, Agriculture and Food Executive Agency (Chafea), acting under powers delegated by the Commission of the European Union (https://ec. europa.eu/chafea/index_en.htm). The funders had no role in study design, data collection and analysis, decision to publish, or preparation of the manuscript.

**Competing interests:** The authors have declared that no competing interests exist.

more screening and a change in testing performance. A country variable representing the proportion of men screened for asymptomatic infection was positively associated with reporting symptomatic gonorrhoea/chlamydia, but not syphilis.

## Discussion/Conclusion

The positive association of country-level screening rates with the proportion of symptomatic infections with gonorrhoea/chlamydia may indicate a paradoxical effect of screening on incidence of symptomatic infections. Treatment of asymptomatic men might render them more susceptible to new infections, while spontaneous clearance may result in reduced susceptibility. Before expanding screening programmes, evidence of the effects of screening and treatment is warranted.

## Introduction

Increasing diagnoses of sexually transmitted infections (STIs), particularly gonorrhoea, chlamydia and syphilis among men who have sex with men (MSM) have been reported from many European and high-income countries in recent years [1–4]. Reasons for these increases are not fully understood and likely differ between gonorrhoea and chlamydia on the one hand and syphilis on the other hand.

Increased numbers of reported gonorrhoea and chlamydia infections may partly be explained by an increase in testing of extra-genital sites, an increased testing frequency, and a switch to nucleic acid amplification tests (NAATs) [5–7]. Screening extra-genital sites for gonorrhoea and chlamydia among asymptomatic MSM has been increasingly adopted in European countries. Recent analyses have shown an increase of anal swabbing as part of STI-screening in all Northern, Western, and Southern, as well as in some Eastern European countries, possibly accounting for a larger increase of the number of reported infections [8]. However, STI-testing policies and practices show large variation across Europe [8], not only between countries, but also within countries and between different service providers.

Declining condom use may be an important behavioural factor contributing to increased STI transmission. However, the relative importance of different transmission routes of gonorrhoea and chlamydia remains unclear: while the possibility of genito-rectal, recto-genital, genito-oral and oro-genital transmission is uncontested, their relative contribution and the efficiency of oro-oral and oro-rectal transmission is disputed [9–11]. The insufficient knowledge about the relative contribution of different transmission routes poses a challenge for analysing the role of behaviour changes for the increased number of diagnoses. While declining condom use for anal intercourse is likely to impact genito-rectal and recto-genital transmission, there is few data available on changes in sexual practice and on the frequency of transmission of infection through oro-genital, oro-rectal or oro-oral contacts [12–14].

It is noteworthy that the increasing adoption of screening for gonorrhoea and chlamydia among MSM was based on scarce evidence: though there is solid evidence that many asymptomatic infections with gonorrhoea and chlamydia can be detected in extra-genital sites in sexually active MSM [15,16], emerging evidence shows that asymptomatic anal and pharyngeal infections clear spontaneously within approximately 3 months without serious or long-term sequelae [17,18]. In addition, there is no published evidence of individual or public health

benefits from screening programmes for asymptomatic gonorrhoea and chlamydia infections among MSM [19].

For syphilis testing, no major changes of test sensitivity occurred that would account for the increased number of diagnoses. Instead, the increased frequency of blood tests and the growing number of HIV and syphilis tests among MSM may have contributed to raising number of reported diagnoses. Increases in syphilis diagnoses have mostly been attributed to behaviour changes such as increasing partner numbers and declining condom use for anal intercourse due to growing awareness that undetectable (HIV viral load) equals untransmissible (U = U), primarily among MSM diagnosed with HIV, but to a lesser extent also among MSM testing negative for HIV, and increased HIV-serosorting based on current HIV test results [20–23]. The introduction and growing availability of HIV pre-exposure prophylaxis (PrEP) for HIV prevention may have further accelerated the declining trend in condom use [21]. Available data suggest a higher proportion of syphilis infections among MSM diagnosed with HIV compared to gonorrhoea and chlamydia infections, but a lower proportion compared to infections with hepatitis C virus and lymphogranuloma venereum [24–29].

In this analysis we use data on self-reported diagnoses of bacterial STIs among MSM collected in two large pan-European online surveys for MSM in 2010 and 2017 to explore the following questions:

1. Was the proportion of respondents reporting diagnosed bacterial STIs in the previous twelve months higher in 2017 than in 2010 (even when controlling for sample composition)?

2. Was the proportion of respondents reporting condomless anal sex in the previous twelve months higher in 2017 than in 2010?

3. Was the proportion of respondents reporting a recent STI screen larger in 2017 than in 2010?

4. Do increases of condomless anal intercourse fully explain the increase of diagnoses of bacterial STIs between the two surveys?

5. Do differences between national screening practices explain the remaining differences in STI diagnoses?

## Methods

### Recruitment and questionnaires

The detailed methods of the European MSM Internet Surveys in 2010 and 2017 (EMIS-2010, EMIS-2017) have been reported elsewhere [30]. In summary, EMIS-2010 and EMIS-2017 were multi-language, internet-based, self-completion surveys for men-who-have-sex-with-men living in Europe. The European MSM Internet Surveys in 2010 and 2017 collected data about sexual behaviours, precautionary behaviours related to HIV, self-assessed STI-testing behaviours including the recency of the last test, testing performance and various self-reported STI diagnoses, including gonorrhoea, chlamydia and syphilis, and collected information whether symptoms were present at the last STI test. The EMIS-2017 questionnaire was based on the version of 2010, and only questions identical in both questionnaires are used for this analysis. Both surveys were available in 25, and respectively, 33 languages across 46 countries. Participants were recruited through trans-national dating apps (PlanetRomeo, Grindr and Hornet accounted for 62% of participants to both surveys collectively, other dating platforms and apps for another 16%), through Facebook, Twitter, Instagram (3%) and through a variety of local

online promotion, mostly through website banners (17%). No financial incentives were given to participants. No personal identifying information (including IP addresses) was collected. Further background information, including all 33 language versions of the questionnaire, is available at www.emis2017.eu. Ethics approval was granted by the Research Ethics Committee of the University of Portsmouth for EMIS-2010 (REC application number 08/09:21), and by the Ethics Committee of the London School of Hygiene and Tropical Medicine for EMIS-2017 (reference 14421/RR/8805).

In terms of recruitment, EMIS-2017 could not replicate the approach used in 2010, since smart phone apps which were the main recruitment source for EMIS-2017 did not yet play a role in 2010 [30].

## Dependent variables

**Primary outcomes.**   All men were asked 'Have you ever been diagnosed with syphilis?' Men who answered yes, were asked 'When were you last diagnosed with syphilis?' and offered a scale to indicate how recently this had been. Identical questions were asked for 'gonorrhoea' and 'chlamydia or LGV'. We grouped syphilis diagnosed in the past 24 hours, seven days, four weeks, six months and twelve months as 'syphilis diagnosed in the past twelve months'. Gonorrhoea and Chlamydia/LGV were grouped accordingly, if either of the two were reported, as 'gonorrhoea/chlamydia diagnosed in the past twelve months'. An exploratory analysis had shown broad similarity regarding the factors associated with gonorrhoea and chlamydia, and the effect sizes of these factors, so we decided to combine gonorrhoea and chlamydia to streamline the presentation of the results and named them gonorrhoea/chlamydia. This also reflects the fact that both STIs are typically tested for in combination, and that respondents might have difficulties distinguishing between the two.

**Secondary outcomes.**   The question whether symptoms were present at the last STI test within the past twelve months allowed to classify self-reported STI diagnoses as *symptomatic* if the diagnosis was made at the last STI test and symptoms were reported at the time of the last STI test, and as *asymptomatic* if no symptoms were reported. If more than one infection was diagnosed at the last STI test and symptoms were reported, they were rendered unclassifiable if syphilis as well as gonorrhoea and/or chlamydia had been diagnosed, as it was not clear which of the diagnosed STIs was symptomatic, if only gonorrhoea and/or chlamydia had been diagnosed, both were classified as symptomatic.

The analyses of asymptomatic STI diagnoses were restricted to men who received an STI testing procedure that would allow the respective diagnosis. A diagnosis of gonorrhoea and chlamydia required that either a genital or a rectal specimen had been collected (urine, urethral swab, rectal swab) as part of STI-testing in the previous twelve months. The collection of pharyngeal swabs had not been queried. A diagnosis of syphilis required a blood test.

This study thus focuses on six independent outcomes: any self-reported diagnoses, diagnoses classified *symptomatic*, diagnoses classified *asymptomatic*, each separated for syphilis and gonorrhoea/chlamydia.

## Independent variables

The survey wave, spanning more than 7 years, was included as a binary variable.

**Survey artefacts.**   The wording for the French translation for STI diagnoses was slightly changed for 2017. After discussion with the multi-national translation team for French, the new wording, while technically correct, may have been misunderstood by some French-speaking respondents. Consequently, the questions on diagnosed syphilis, gonorrhoea, and chlamydia may have been understood by some men as having undergone a test rather than having

a positive test result. This problem affects all three countries with large sub-samples using the French version, notably France, Belgium and Switzerland, in descending order for decreasing proportions of French speakers, as reflected by the disproportionate increase of self-reported STI diagnoses in these countries. To control for a potential overestimation of STI diagnoses in French questionnaires, a binary language variable (French–not French) was constructed. We further controlled for major discrepancies (discrepant answers for age, steady partners, or non-steady partners), using a binary variable. Such discrepancies occur when respondents either give random answers or always select *e.g.* the first response options.

**Sample composition.** In the multivariable regression models age was included as continuous variable, HIV diagnosis as a binary variable, settlement size as an ordinal variable with five categories.

**Testing behaviour.** As a proxy for testing frequency, the recency of the last STI test was categorised as 'no screening', 'screening 6–12 months ago', '1–6 months ago', 'within the last 4 weeks'. Respondents reporting symptoms at the last STI test were included in 'no screening'. Testing behaviour is considered as individual behaviour, irrespective of any given testing procedures.

**Sexual behaviour.** As the number of sexual partners is the major determinant for STI transmission, we included an ordinal variable for the number of overall sexual partners in the previous twelve months: None or one, 2–4, 5–7, 8–10, 11–20, >20. Based on the *last sexual encounter* with one or more non-steady partners, we constructed an ordinal variable combined for anal intercourse and condom use: No non–steady partner in the previous twelve months, no anal intercourse (AI) with the last non-steady partner, no condomless AI, some condomless AI, only condomless AI.

**Country-level screening rates.** To capture STI testing practices and performance two additional country-level variables were constructed and categorized by quartiles: a) being screened for syphilis, defined as reporting no symptoms at last test and reporting a blood-based test as part of STI-testing in the previous twelve months; b) being screened for gonorrhoea/chlamydia, defined as reporting no symptoms at last test and reporting a test based on a genital specimen (urine or urethral swab) <u>and</u> an anal swab as part of STI-testing in the previous twelve months. Subtracted from numerator and denominator were men reporting an STI diagnosis that was *unclassifiable* or classified *symptomatic*. This variable was used for both primary outcomes and for STIs classified *symptomatic*–this variable was not applicable for STIs classified *asymptomatic*, as these were already restricted to respondents screened for STIs. Instead, for gonorrhoea/chlamydia classified as *asymptomatic*, we further controlled for the reception of an anal swab, as it reflects the testing practices of the testing centres rather than the individual's testing behaviour.

## Statistics

For continuous variables mean, Standard Deviation (SD), median and interquartile range (IQR) was used. For nominal variables count and percentages were used.

Based on theoretical assumptions a list of variables, potentially associated with the dependent variables, was developed for each outcome variable. In order to identify the list of significantly associated variables with the dependent variables a bivariate approach was used and a two-level multilevel logistic regression model with a random intercept at country level. The random component accounts for the hierarchical nature of the data. Analyses were carried out on all available cases of each survey wave.

A model was developed for each dependent variable. The first step to building a model was to enter sequentially those individual independent variables that were statistically significantly

associated with the dependent variable for each model (based on bivariate analysis). The variable indicating the survey wave was included at the beginning of the modelling and retained for all successive model building steps. Age was also included as confounder to be controlled for as potentially associated with the outcome variables. Variables from the significantly associated pool were then included sequentially in the multivariate analysis. The variables were added to the null model one by one retaining those variables significant at p<0.05. The final models where then estimated with the pool of significantly associated variables and a random slope was also included for the survey year variable. The likelihood ratio test was used to compare the new model with the nested model to establish the model improvement. For all statistical tests, significance was indicated by p<0.05. The final model estimated the adjusted odds ratios (aORs) and the corresponding 95% confidence interval (95% CI) for factors associated with the dependent variable. Analyses were carried out using Stata® Version 15.1 (College Station, TX: StataCorp LP).

## Results

### Changes in STI diagnoses

Responses to both, STI diagnosis questions and sexual behaviours, were contributed by 156,018 participants in 2010 and 125,837 participants in 2017 from 46 countries in Europe. The final sample sizes for the multivariate models were smaller due to missing responses, mainly on STI screening. Overall diagnoses with syphilis increased between 2010 and 2017 across countries by 83%, symptomatic infections with syphilis increased by 28%, overall infections with gonorrhoea/chlamydia increased by 76%, symptomatic infections by 31%, with considerable variation between countries. Asymptomatic infections increased by 145% for syphilis, and 142% for gonorrhoea/chlamydia. The proportion of unclassifiable diagnoses was 50–52% for the two STI types in 2010, and 53–55% in 2017 (see S1 and S2 Tables).

Fig 1 shows a panel of 6 country-level scatter plots: (a), (b), and (c) show the correlation between overall syphilis diagnosis rates in the previous twelve months by 2010 and 2017 survey waves, diagnosis rates for classified symptomatic syphilis, and diagnosis rates for classified asymptomatic syphilis; (d), (e), and (f) show the respective correlations for the combined gonorrhoea and chlamydia diagnosis rates. All countries plotted above the line of equality have experienced increased diagnosis rates between 2010 and 2017. As the figure demonstrates, almost all countries have experienced increased STI diagnosis rates to varying degrees.

The supplementary S1 and S2 Tables present the overall numbers of participants of the two survey waves by country, and number and proportions of all self-reported infections with syphilis, and/or gonorrhoea/chlamydia in 2010 and 2017, of the subset of infections classified symptomatic, and number and proportions of infections classified asymptomatic among those who had been screened. The two tables also present the numbers and proportions of participants of the two survey waves that have been screened with a blood-based STI test, adequate for a serological diagnosis of syphilis, and with a urine-based STI test or genital swab and an anal swab that can be used for the diagnosis of genital or rectal infections with gonorrhoea or chlamydia.

### Changes in STI-testing

Comprehensive screening rates for gonorrhoea/chlamydia increased across countries by 103%, and for syphilis by 54%; considerable variation between countries was observed.

Fig 2 shows a panel of two country-level scatter plots: a) the proportion of participants screened with a blood-based STI test in the previous twelve months, b) the proportion of

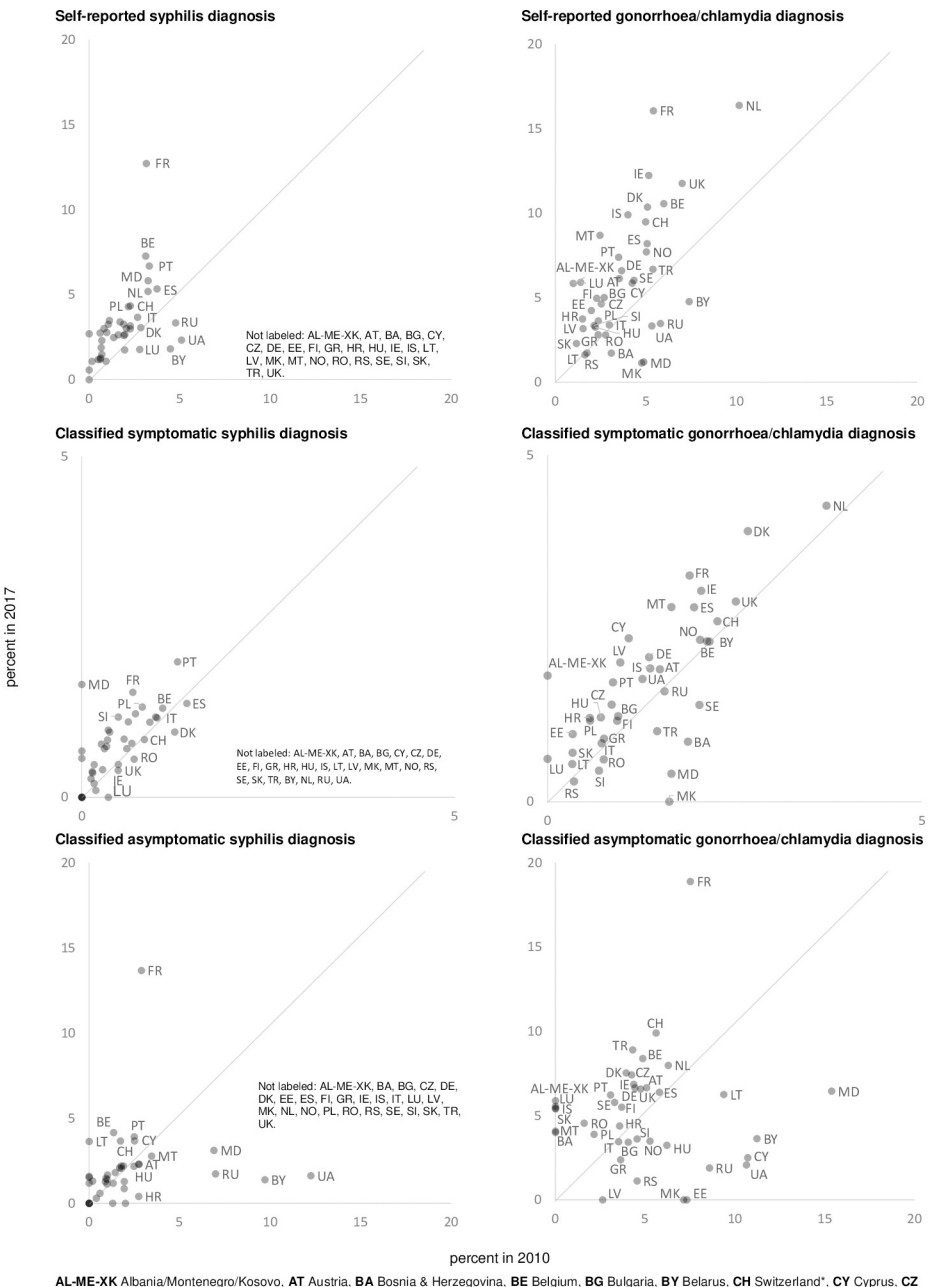

**Fig 1. Main outcome variables.** Panel of 6 country-level scatter plots: Syphilis diagnosis; Gonorrhoea/Chlamydia diagnosis; Classified symptomatic syphilis diagnosis; Classified symptomatic gonorrhoea/chlamydia diagnosis; Classified asymptomatic syphilis diagnosis; Classified asymptomatic gonorrhoea/chlamydia diagnosis.

participants screened with a urine-based STI test or genital swab **and** an anal swab in the previous twelve months. As the figure demonstrates, in almost all countries the proportions of men screened with a blood-based STI test and the combination of a urine-based test or genital swab and an anal swab have increased to different extents.

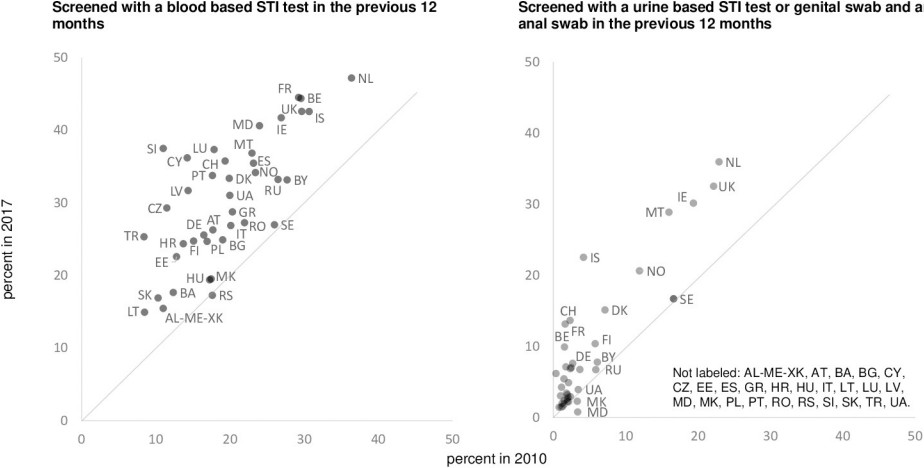

**Fig 2. Explanatory variables.** Panel of 2 country-level scatter plots (interventions): Screened with a blood based STI test in the previous 12 months; Screened with a urine based STI test or genital swab <u>and</u> an anal swab in the previous 12 months.

## Changes in sexual behaviour

The proportion of respondents reporting more than ten sex partners in the previous twelve months increased by 10%, the proportion reporting any condomless anal intercourse with the last non-steady partner increased by 62%, and the proportion reporting an STI screen within the last 6 months increased by 63%.

Fig 3 shows a panel of three country-level scatter plots presenting changes in behaviour between the two survey waves: a) the proportion of participants reporting more than ten sexual partners in the previous twelve months; b) the proportion of participants reporting condomless anal intercourse with the last non-steady partner; and c) the proportion of participants reporting an STI screen in the previous six months, used as a proxy for the frequency of STI testing. The respective data for these three country level scatter plots are also presented in supplementary S3 Table. Fig 3 demonstrates increasing partner numbers, declining condom use, and increasing STI test frequency between the two survey waves in 2010 and 2017.

## Multilevel multivariate regression models, syphilis

Tables 1 and 2 present the results of bivariate and multilevel multivariate regression (MMR) analyses for the following outcomes: all, classified symptomatic, and classified asymptomatic syphilis and gonorrhoea/chlamydia.

The unadjusted odds ratio for the survey wave difference between 2010 and 2017 indicates the–significant–differences in diagnosis rates between the two survey waves for all six outcome variables. The adjusted odds ratios in Tables 1A and 2A show how much these differences between the survey waves are explained by survey artefacts, sample composition, sexual and testing behaviours. The adjusted odds ratios in Tables 1B and 2B all become insignificant, showing that the inclusion of further country level intervention parameters in the multivariate models largely explains the remaining differences between the two survey waves.

**For all self-reported syphilis diagnoses in the previous twelve months,** the MMR model showed no significant difference for the two survey waves after adjustment for all other factors included in the model. Respondents filling in the 2017 French language version had a higher

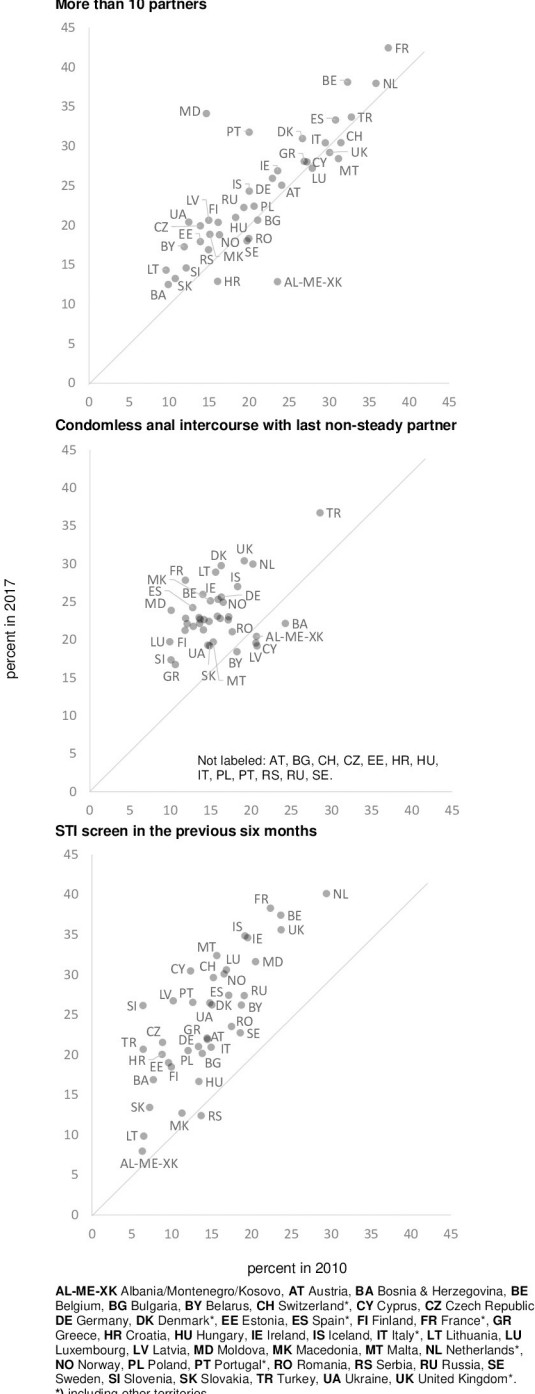

**Fig 3. Explanatory variables.** Panel of 3 country-level scatter plots (behaviour): More than 10 partners; Condomless anal intercourse with last non-steady partner; STI screen in the previous six months.

probability to report a syphilis diagnosis in the previous twelve months, as had respondents whose questionnaires contained any of the aforementioned logical discrepancies. Age had no impact on syphilis diagnosis. The probability to report a syphilis diagnosis slightly increased with increasing settlement size. Living with diagnosed HIV was strongly associated with a

**Table 1. a. Univariable and multivariable regression analysis, main model: Individual-level associations with syphilis diagnosis. b. Univariable and multivariable regression analysis, comprehensive model: Individual-level associations with syphilis diagnosis including impact of country-level[1] screening rates on syphilis diagnosis (intervention practices).**

| Fixed part | | Full sample, excluding only MSM with missing answers | | | | | | | | Sub-sample of MSM screened[2] | | | |
| --- | --- | --- | --- | --- | --- | --- | --- | --- | --- | --- | --- | --- | --- |
| | | Diagnosis of syphilis, any (N = 262,995) | | | | Diagnosis of syphilis, classified symptomatic (N = 270,164) | | | | Diagnosis of syphilis, classified asymptomatic (N = 71,462) | | | |
| | | OR | 95%-CI | AOR | 95%-CI | OR | 95%-CI | AOR | 95%-CI | OR | 95%-CI | AOR | 95%-CI |
| **Time** | | | | | | | | | | | | | |
| Survey year | 2010 | Ref. | | Ref. | | Ref. | | Ref. | | Ref. | | Ref. | |
| | 2017 | **1.82** | 1.75–1.90 | **1.17** | 1.01–1.35 | **1.30** | 1.20–1.41 | 1.02 | 0.90–1.16 | **1.54** | 1.41–1.69 | 0.81 | 0.63–1.05 |
| **Survey artefacts** | | | | | | | | | | | | | |
| French translation | Not used 2010/2017 | Ref. | | Ref. | | n.a. | | Ref. | 1.00–1.00 | n.a. | | Ref. | |
| | Used in 2010 | **1.59** | 1.41–1.80 | 1.10 | 0.90–1.36 | 1.10 | 0.85–1.43 | 0.92 | 0.67–1.26 | **3.21** | 2.52–4.08 | 1.43 | 0.93–2.22 |
| | Used in 2017 | | | **2.08** | 1.75–2.46 | | | 1.24 | 0.90–1.72 | | | **4.54** | 3.35–6.16 |
| Discrepant data | No | Ref. | | Ref. | | Ref. | | Ref. | 1.00–1.00 | Ref. | | Ref. | |
| | Yes | **1.22** | 1.16–1.29 | **1.22** | 1.14–1.30 | **1.17** | 1.04–1.30 | 1.07 | 0.95–1.21 | **1.23** | 1.09–1.39 | **1.29** | 1.14–1.47 |
| **Sample composition** | | | | | | | | | | | | | |
| Age | Per year | **1.02** | 1.01–1.02 | 1.00 | 1.00–1.00 | **1.01** | 1.01–1.02 | 1.00 | 1.00–1.00 | **1.01** | 1.01–1.02 | 1.00 | 1.00–1.01 |
| Settlement size | Village/countryside (<10,000) | Ref. | | Ref. | | Ref. | | Ref. | 1.00–1.00 | Ref. | | Ref. | |
| (inhabitants) | Small town (10,000–99,999) | 1.06 | 0.97–1.15 | 1.07 | 0.97–1.17 | 1.01 | 0.86–1.18 | 1.00 | 0.85–1.18 | 0.91 | 0.77–1.08 | 0.93 | 0.78–1.11 |
| | Medium town (100,000–499,999) | **1.23** | 1.13–1.33 | **1.15** | 1.05–1.26 | **1.22** | 1.04–1.42 | 1.12 | 0.95–1.31 | **0.77** | 0.65–0.91 | **0.82** | 0.69–0.97 |
| | Big city (500,000–999,999) | **1.58** | 1.46–1.72 | **1.27** | 1.16–1.39 | **1.50** | 1.27–1.76 | 1.17 | 0.99–1.38 | **0.80** | 0.67–0.95 | **0.82** | 0.69–0.98 |
| | Very big city (≥1 million) | **1.72** | 1.59–1.86 | **1.21** | 1.11–1.31 | **1.50** | 1.30–1.74 | 1.04 | 0.89–1.21 | **0.71** | 0.61–0.82 | **0.73** | 0.62–0.85 |
| Diagnosed HIV | No | Ref. | | Ref. | | Ref. | | Ref. | 1.00–1.00 | Ref. | | Ref. | |
| | Yes | **6.52** | 6.24–6.81 | **3.97** | 3.77–4.18 | **4.98** | 4.57–5.44 | **3.17** | 2.88–3.50 | **1.94** | 1.76–2.15 | **1.76** | 1.57–1.96 |
| **Testing behaviour** | | | | | | | | | | | | | |
| STI screen | No screening | Ref. | | Ref. | | | | | | | | | |
| | 6–12 months ago | **0.89** | 0.81–0.97 | **0.66** | 0.61–0.73 | | | | | | | | |
| | 1–6 months ago | **2.22** | 2.11–2.34 | **1.14** | 1.08–1.21 | | | | | | | | |
| | Within the last 4 weeks | **4.00** | 3.77–4.24 | **1.72** | 1.61–1.84 | | | | | | | | |
| **Sexual behaviour** | | | | | | | | | | | | | |
| Number of sex partners, | None or one | Ref. | | Ref. | | Ref. | | Ref. | 1.00–1.00 | Ref. | | Ref. | |
| previous 12 months | 2–4 | **1.43** | 1.31–1.55 | **1.49** | 1.31–1.69 | **1.60** | 1.34–1.90 | **1.54** | 1.19–1.99 | 1.10 | 0.93–1.30 | 1.07 | 0.83–1.38 |
| | 5–7 | **2.18** | 2.00–2.38 | **2.18** | 1.90–2.51 | **2.66** | 2.23–3.17 | **2.59** | 1.96–3.42 | 1.16 | 0.97–1.39 | 1.17 | 0.89–1.54 |
| | 8–10 | **2.93** | 2.66–3.22 | **2.67** | 2.31–3.09 | **3.65** | 3.03–4.41 | **3.29** | 2.47–4.37 | **1.45** | 1.20–1.76 | **1.39** | 1.04–1.84 |
| | 11–20 | **3.80** | 3.52–4.11 | **3.21** | 2.80–3.68 | **4.97** | 4.24–5.82 | **4.14** | 3.16–5.43 | **1.34** | 1.14–1.58 | 1.29 | 0.98–1.69 |
| | >20 | **8.20** | 7.62–8.81 | **5.40** | 4.72–6.17 | **9.06** | 7.8–10.53 | **6.37** | 4.88–8.32 | **1.85** | 1.59–2.15 | **1.62** | 1.24–2.10 |
| Anal intercourse, condom use with last non–steady partner | No non–steady partner | Ref. | | Ref. | | Ref. | | Ref. | 1.00–1.00 | Ref. | | Ref. | |
| | No anal intercourse | **1.92** | 1.77–2.09 | **0.70** | 0.62–0.80 | **2.60** | 2.21–3.05 | 0.89 | 0.69–1.15 | 1.01 | 0.86–1.20 | **0.77** | 0.59–1.00 |
| | No condomless anal int. | **2.35** | 2.19–2.53 | **0.82** | 0.72–0.93 | **2.91** | 2.51–3.37 | 0.96 | 0.74–1.23 | 1.14 | 0.99–1.32 | 0.87 | 0.68–1.12 |
| | Some condomless anal int. | **5.75** | 5.15–6.41 | **1.38** | 1.18–1.62 | **6.14** | 4.93–7.66 | **1.67** | 1.23–2.27 | **2.09** | 1.65–2.64 | 1.28 | 0.93–1.76 |
| | Only condomless anal int. | **6.20** | 5.78–6.65 | **1.41** | 1.24–1.60 | **6.61** | 5.72–7.65 | **1.61** | 1.25–2.08 | **2.17** | 1.87–2.52 | **1.32** | 1.03–1.70 |
| **Random part** | | $\sigma^2$ | 95%-CI | $\sigma^2$ | 95%-CI | $\sigma^2$ | 95%-CI | $\sigma^2$ | 95%-CI | $\sigma^2$ | 95%-CI | $\sigma^2$ | 95%-CI |
| Survey year | Random Slope | | | 0.12 | 0.06–0.25 | | | 0.02 | 0.00–0.10 | | | 0.28 | 0.12–0.63 |
| 46 countries[1] | Random Intercept | | | 0.23 | 0.13–0.41 | | | 0.19 | 0.10–0.37 | | | 0.50 | 0.28–0.91 |
| Likelihood-ratio test[3] | df, p | | | 25 | p < 0.001 | | | 22 | p < 0.001 | | | 22 | p < 0.001 1<0.001 = xxx |
| **Time** | | | | | | | | | | | | | |

*(Continued)*

**Table 1.** (Continued)

| | | Full sample, excluding only MSM with missing answers | | | | | | | | Sub-sample of MSM screened[2] | | | |
| --- | --- | --- | --- | --- | --- | --- | --- | --- | --- | --- | --- | --- | --- |
| | | Diagnosis of syphilis, any (N = 262,995) | | | | Diagnosis of syphilis, classified symptomatic (N = 270,164) | | | | Diagnosis of syphilis, classified asymptomatic (N = 71,462) | | | |
| **Fixed part** | | OR | 95%-CI | AOR | 95%-CI | OR | 95%-CI | AOR | 95%-CI | OR | 95%-CI | AOR | 95%-CI |
| Survey year | 2010 | Ref. | | Ref. | | Ref. | | Ref. | | Ref. | | Ref. | |
| | 2017 | **1.82** | 1.75–1.90 | 1.09 | 0.87–1.38 | **1.30** | 1.20–1.41 | 1.13 | 0.91–1.40 | **1.54** | 1.41–1.69 | 0.81 | 0.63–1.05 |
| **Survey artefacts** | | | | | | | | | | | | | |
| French translation | Not used 2010/2017 | Ref. | | Ref. | | n.a. | | Ref. | 1.00–1.00 | n.a. | | Ref. | |
| | Used in 2010 | **1.59** | 1.41–1.80 | 1.08 | 0.88–1.34 | 1.10 | 0.85–1.43 | 0.89 | 0.65–1.21 | **3.21** | 2.52–4.08 | 1.43 | 0.93–2.22 |
| | Used in 2017 | | | **2.08** | 1.75–2.46 | | | 1.32 | 0.98–1.78 | | | **4.54** | 3.35–6.16 |
| Discrepant data | No | Ref. | | Ref. | | Ref. | | Ref. | 1.00–1.00 | Ref. | | Ref. | |
| | Yes | **1.22** | 1.16–1.29 | **1.22** | 1.14–1.30 | **1.17** | 1.04–1.30 | 1.07 | 0.95–1.21 | **1.23** | 1.09–1.39 | **1.29** | 1.14–1.47 |
| **Sample composition** | | | | | | | | | | | | | |
| Age | Per year | **1.02** | 1.01–1.02 | 1.00 | 1.00–1.00 | **1.01** | 1.01–1.02 | 1.00 | 1.00–1.00 | **1.01** | 1.01–1.02 | 1.00 | 1.00–1.01 |
| Settlement size | Village/countryside (<10,000) | Ref. | | Ref. | | Ref. | | Ref. | 1.00–1.00 | Ref. | | Ref. | |
| (inhabitants) | Small town (10,000–99,999) | 1.06 | 0.97–1.15 | 1.07 | 0.98–1.17 | 1.01 | 0.86–1.18 | 1.00 | 0.85–1.18 | 0.91 | 0.77–1.08 | 0.93 | 0.78–1.11 |
| | Medium town (100,000–499,999) | **1.23** | 1.13–1.33 | **1.15** | 1.06–1.26 | **1.22** | 1.04–1.42 | 1.12 | 0.95–1.31 | **0.77** | 0.65–0.91 | **0.82** | 0.69–0.97 |
| | Big city (500,000–999,999) | **1.58** | 1.46–1.72 | **1.27** | 1.16–1.39 | **1.50** | 1.27–1.76 | 1.18 | 1.00–1.39 | **0.80** | 0.67–0.95 | **0.82** | 0.69–0.98 |
| | Very big city (≥1 million) | **1.72** | 1.59–1.86 | **1.21** | 1.12–1.32 | **1.50** | 1.30–1.74 | 1.04 | 0.89–1.21 | **0.71** | 0.61–0.82 | **0.73** | 0.62–0.85 |
| Diagnosed HIV | No | Ref. | | Ref. | | Ref. | | Ref. | 1.00–1.00 | Ref. | | Ref. | |
| | Yes | **6.52** | 6.24–6.81 | **3.97** | 3.77–4.19 | **4.98** | 4.57–5.44 | **3.18** | 2.89–3.51 | **1.94** | 1.76–2.15 | **1.76** | 1.57–1.96 |
| **Testing behaviour** | | | | | | | | | | | | | |
| STI screen | No screening | Ref. | | Ref. | | | | | | | | | |
| | 6–12 months ago | **0.89** | 0.81–0.97 | **0.66** | 0.61–0.72 | | | | | | | | |
| | 1–6 months ago | **2.22** | 2.11–2.34 | **1.14** | 1.08–1.21 | | | | | | | | |
| | Within the last 4 weeks | **4.00** | 3.77–4.24 | **1.72** | 1.61–1.84 | | | | | | | | |
| **Sexual behaviour** | | | | | | | | | | | | | |
| Number of sex partners, | None or one | Ref. | | Ref. | | Ref. | | Ref. | 1.00–1.00 | Ref. | | Ref. | |
| previous 12 months | 2–4 | **1.43** | 1.31–1.55 | **1.49** | 1.31–1.69 | **1.60** | 1.34–1.90 | **1.54** | 1.18–1.99 | 1.10 | 0.93–1.30 | 1.07 | 0.83–1.38 |
| | 5–7 | **2.18** | 2.00–2.38 | **2.18** | 1.90–2.51 | **2.66** | 2.23–3.17 | **2.59** | 1.96–3.42 | 1.16 | 0.97–1.39 | 1.17 | 0.89–1.54 |
| | 8–10 | **2.93** | 2.66–3.22 | **2.68** | 2.31–3.10 | **3.65** | 3.03–4.41 | **3.28** | 2.47–4.37 | **1.45** | 1.20–1.76 | **1.39** | 1.04–1.84 |
| | 11–20 | **3.80** | 3.52–4.11 | **3.21** | 2.80–3.68 | **4.97** | 4.24–5.82 | **4.14** | 3.16–5.43 | **1.34** | 1.14–1.58 | 1.29 | 0.98–1.69 |
| | >20 | **8.20** | 7.62–8.81 | **5.40** | 4.72–6.18 | **9.06** | 7.8–10.53 | **6.37** | 4.88–8.33 | **1.85** | 1.59–2.15 | **1.62** | 1.24–2.10 |
| Anal intercourse, condom use with last non–steady partner | No non–steady partner | Ref. | | Ref. | | Ref. | | Ref. | 1.00–1.00 | Ref. | | Ref. | |
| | No anal intercourse | **1.92** | 1.77–2.09 | **0.70** | 0.62–0.80 | **2.60** | 2.21–3.05 | 0.89 | 0.69–1.15 | 1.01 | 0.86–1.20 | **0.77** | 0.59–1.00 |
| | No condomless anal int. | **2.35** | 2.19–2.53 | **0.82** | 0.72–0.93 | **2.91** | 2.51–3.37 | 0.96 | 0.75–1.23 | 1.14 | 0.99–1.32 | 0.87 | 0.68–1.12 |
| | Some condomless anal int. | **5.75** | 5.15–6.41 | **1.38** | 1.18–1.61 | **6.14** | 4.93–7.66 | **1.68** | 1.24–2.28 | **2.09** | 1.65–2.64 | 1.28 | 0.93–1.76 |
| | Only condomless anal int. | **6.20** | 5.78–6.65 | **1.41** | 1.24–1.60 | **6.61** | 5.72–7.65 | **1.61** | 1.26–2.08 | **2.17** | 1.87–2.52 | **1.32** | 1.03–1.70 |
| **Country-level proportion of survey participants screened in 2010 and 2017** | | | | | | | | | | | | | |
| **In the full sample, screened with a blood-based STI test[4]** | | | | | | | | | | | | | |
| <17.54% | Quartile 1 | Ref. | | | | Ref. | | | | | | | |
| 17.54–24.54% | Quartile 2 | **1.67** | 1.50–1.86 | **1.20** | 1.06–1.36 | **1.33** | 1.01–1.75 | 1.12 | 0.91–1.39 | | | | |
| 24.55–31.37% | Quartile 3 | **1.74** | 1.61–1.89 | **1.23** | 1.06–1.42 | **1.41** | 1.05–1.91 | 0.93 | 0.73–1.20 | | | | |
| 31.38–47.20% | Quartile 4 | **3.43** | 3.10–3.79 | 1.21 | 0.96–1.52 | 1.37 | 0.90–2.08 | 0.86 | 0.59–1.25 | | | | |
| **Random part** | | $\sigma^2$ | 95%-CI | $\sigma^2$ | 95%-CI | $\sigma^2$ | 95%-CI | $\sigma^2$ | 95%-CI | $\sigma^2$ | 95%-CI | $\sigma^2$ | 95%-CI |
| Survey year | Random Slope | | | 0.14 | 0.07–0.29 | | | n.a. | | | | 0.28 | 0.12–0.63 |
| 46 countries[1] | Random Intercept | | | 0.18 | 0.10–0.32 | | | 0.20 | 0.10–0.40 | | | 0.50 | 0.28–0.91 |

(Continued)

**Table 1.** (Continued)

| Fixed part | | Full sample, excluding only MSM with missing answers | | | | | | | | Sub-sample of MSM screened[2] | | | |
| | | Diagnosis of syphilis, any (N = 262,995) | | | | Diagnosis of syphilis, classified symptomatic (N = 270,164) | | | | Diagnosis of syphilis, classified asymptomatic (N = 71,462) | | | |
| | | OR | 95%-CI | AOR | 95%-CI | OR | 95%-CI | AOR | 95%-CI | OR | 95%-CI | AOR | 95%-CI |
|---|---|---|---|---|---|---|---|---|---|---|---|---|---|
| Likelihood-ratio test[5] | df, p | | | 28 | p = 0.01 | | | 24 | p = 1.0 | | | 22 | n.a. |

[1] This study includes 46 countries, with four European microstates included in neighbouring (Andorra, Liechtenstein) or surrounding (Monaco, San Marino) countries, and with Albania, Montenegro and Kosovo merged to form a region; this results in 40 country-like entities included in the random part of the model.

[2] Includes only respondents screened with a blood-based STI test.

[3] comparing multilevel model with linear regression model.

[4] grouped by quartiles.

[5] comparing model in Table 1A with model in Table 1B. n.a. = not applicable.

syphilis diagnosis. Increased frequency of STI-screening was associated with a higher probability of a syphilis diagnosis. Increasing numbers of sexual partners and declining consistency of condom use were both associated with higher odds of a syphilis diagnosis. Men who reported no anal intercourse or only condom-protected anal intercourse with the last non-steady partner had a lower likelihood for a syphilis diagnosis than men who had no non-steady partners in the previous twelve months. Living in a country with screening rates for blood-based STI tests in the second or third quartile of country-level screening rates was associated with a higher probability of a syphilis diagnosis.

**For syphilis infections classified as symptomatic**, the associations and their direction in the MMR model were similar to the overall syphilis model, with survey artefacts and sample composition variables becoming non-significant except for diagnosed HIV and sexual behaviour variables showing stronger associations. As shown in Table 1B, the country-level proportions of survey participants screened with blood-based STI tests were not significantly associated with a diagnosis of symptomatic syphilis, and the inclusion of this variable did not improve the model. This variable was only included in the model to allow comparison of the impact of the inclusion of this country-level screening variable on the probability of symptomatic infections between Tables 1B and 2B.

**For syphilis classified as asymptomatic**, after controlling for all other explanatory variables, there was no significant difference between the two survey years; compared with symptomatic syphilis, the probability for asymptomatic syphilis was much higher for the 2017 French language version, and much lower for men diagnosed with HIV. In contrast with symptomatic syphilis, diagnosis probabilities decreased with increasing settlement size, and associations with increasing partner numbers and decreased condom use were much weaker.

## Multilevel multivariate regression models, gonorrhoea/chlamydia

**For all self-reported diagnoses with gonorrhoea/chlamydia in the previous twelve months,** the MMR model in Table 2A still showed a significant difference for the two survey waves after adjustment for all other factors included in the model; however this difference disappears when country-level screening rates are included in the model in Table 2B. Respondents filling in the 2017 French language version had higher probabilities to report a diagnosis of gonorrhoea/chlamydia in the previous twelve months, as had respondents whose questionnaires contained any logical discrepancies, but both aORs were lower for gonorrhoea/chlamydia than for syphilis. Age had a significant impact on diagnosis of gonorrhoea/chlamydia, with a 2% decline per year increase. The probability for reporting a diagnosis of gonorrhoea/chlamydia

**Table 2. a.** Univariable and multivariable regression analysis, main model: individual-level associations with gonorrhoea/chlamydia diagnosis. **b.** Univariable and multivariable regression analysis, main model: individual-level associations with gonorrhoea/chlamydia diagnosis including impact of country-level[1] screening rates on gonorrhoea/chlamydia diagnosis (intervention practices).

| | | Full sample, excluding only MSM with missing answers | | | | | | | | Sub-sample of MSM screened[2] | | | |
| | | Diagnosis of gonorrhoea/chlamydia, any (N = 262,995) | | | | Diagnosis of gonorrhoea/chlamydia, classified symptomatic (N = 266,407) | | | | Diagnosis of gonorrhoea/chlamydia, classified asymptomatic (N = 47,684) | | | |
| **Fixed part** | | OR | 95%-CI | AOR | 95%-CI | OR | 95%-CI | AOR | 95%-CI | OR | 95%-CI | AOR | 95%-CI |
|---|---|---|---|---|---|---|---|---|---|---|---|---|---|
| **Time** | | | | | | | | | | | | | |
| Survey year | 2010 | Ref. | | Ref. | 1 | Ref. | | Ref. | 1.00–1.00 | Ref. | | Ref. | |
| | 2017 | **1.79** | 1.74–1.85 | **1.46** | 1.29–1.65 | **1.34** | 1.27–1.42 | **1.21** | 1.11–1.32 | **1.39** | 1.29–1.50 | 0.98 | 0.81–1.19 |
| **Survey artefacts** | | | | | | | | | | | | | |
| French translation | Not used 2010/2017 | Ref. | . | Ref. | | n.a. | . | Ref. | 1.00–1.00 | n.a. | | Ref. | |
| | Used in 2010 | **1.27** | 1.16–1.40 | 0.86 | 0.74–1.01 | 0.91 | 0.76–1.09 | **0.71** | 0.57–0.89 | **2.87** | 2.38–3.45 | **1.50** | 1.11–2.01 |
| | Used in 2017 | | | **1.57** | 1.38–1.80 | | | 1.00 | 0.81–1.24 | | | **3.15** | 2.46–4.03 |
| Discrepant data | No | Ref. | | Ref. | | Ref. | | Ref. | 1.00–1.00 | Ref. | x | Ref. | |
| | Yes | **1.10** | 1.05–1.15 | **1.06** | 1.01–1.12 | 0.99 | 0.91–1.07 | **0.90** | 0.83–0.99 | 0.98 | 0.88–1.09 | 1.01 | 0.90–1.13 |
| **Sample composition** | | | | | | | | | | | | | |
| Age | Per year | 0.99 | 0.99–1.00 | **0.98** | 0.98–0.98 | **0.99** | 0.99–0.99 | **0.98** | 0.97–0.98 | **0.99** | 0.98–0.99 | **0.98** | 0.98–0.98 |
| Settlement size | Village/countryside (<10,000) | Ref. | | Ref. | | Ref. | | Ref. | 1.00–1.00 | Ref. | | Ref. | |
| (inhabitants) | Small town (10,000–99,999) | **1.15** | 1.08–1.23 | **1.14** | 1.06–1.23 | **1.26** | 1.11–1.43 | **1.22** | 1.07–1.39 | 1.01 | 0.87–1.19 | 1.02 | 0.87–1.20 |
| | Medium town (100,000–499,999) | **1.58** | 1.48–1.68 | **1.43** | 1.33–1.53 | **1.66** | 1.47–1.87 | **1.45** | 1.28–1.64 | **1.17** | 1.01–1.35 | 1.15 | 0.98–1.34 |
| | Big city (500,000–999,999) | **2.02** | 1.89–2.16 | **1.65** | 1.53–1.77 | **2.11** | 1.87–2.38 | **1.69** | 1.49–1.92 | **1.18** | 1.01–1.38 | 1.13 | 0.97–1.33 |
| | Very big city (≥1 million) | **2.56** | 2.41–2.73 | **1.94** | 1.81–2.07 | **2.72** | 2.43–3.04 | **2.06** | 1.83–2.32 | **1.26** | 1.10–1.45 | **1.21** | 1.05–1.40 |
| Diagnosed HIV | No | Ref. | | Ref. | | Ref. | | Ref. | 1.00–1.00 | Ref. | | Ref. | |
| | Yes | **3.08** | 2.96–3.20 | **1.88** | 1.80–1.97 | **2.01** | 1.86–2.17 | **1.37** | 1.26–1.49 | 0.92 | 0.84–1.01 | **0.85** | 0.76–0.94 |
| **Testing behaviour** | | | | | | | | | | | | | |
| STI screen | No screening | Ref. | | Ref. | | | | | | | | | |
| | 6–12 months ago | **0.76** | 0.71–0.81 | **0.56** | 0.52–0.60 | | | | | | | | |
| | 1–6 months ago | **1.90** | 1.83–1.97 | 1.04 | 1.00–1.08 | | | | | | | | |
| | Within the last 4 weeks | **3.41** | 3.25–3.57 | **1.56** | 1.48–1.64 | | | | | | | | |
| **Sexual behaviour** | | | | | | | | | | | | | |
| Number of sex partners, | None or one | Ref. | | Ref. | | Ref. | | Ref. | 1.00–1.00 | Ref. | | Ref. | |
| previous 12 months | 2–4 | **1.71** | 1.60–1.83 | **1.97** | 1.78–2.18 | **1.83** | 1.63–2.06 | **1.88** | 1.58–2.25 | **1.33** | 1.14–1.56 | **1.33** | 1.07–1.67 |
| | 5–7 | **2.90** | 2.71–3.11 | **3.21** | 2.88–3.59 | **3.24** | 2.88–3.64 | **3.23** | 2.67–3.89 | **1.53** | 1.30–1.80 | **1.50** | 1.18–1.92 |
| | 8–10 | **4.03** | 3.74–4.34 | **4.21** | 3.76–4.72 | **4.02** | 3.53–4.58 | **3.88** | 3.19–4.72 | **2.04** | 1.72–2.42 | **1.99** | 1.55–2.56 |
| | 11–20 | **5.75** | 5.40–6.12 | **5.91** | 5.31–6.58 | **5.36** | 4.80–5.99 | **5.13** | 4.27–6.18 | **2.32** | 2.00–2.68 | **2.30** | 1.81–2.91 |
| | >20 | **11.16** | 10.51–11.84 | **9.95** | 8.94–11.1 | **9.12** | 8.20–10.13 | **8.01** | 6.67–9.62 | **3.37** | 2.93–3.87 | **3.29** | 2.61–4.15 |
| Anal intercourse, condom use with last non–steady partner | No non–steady partner | Ref. | | Ref. | | Ref. | | Ref. | 1.00–1.00 | Ref. | | Ref. | |
| | No anal intercourse | **2.66** | 2.50–2.83 | **0.70** | 0.63–0.77 | **2.81** | 2.52–3.14 | 0.85 | 0.71–1.01 | **1.69** | 1.46–1.96 | 0.88 | 0.70–1.10 |
| | No condomless anal int. | **3.43** | 3.25–3.63 | **0.84** | 0.77–0.93 | **3.60** | 3.25–3.98 | 1.03 | 0.87–1.22 | **1.71** | 1.50–1.95 | 0.89 | 0.72–1.10 |
| | Some condomless anal int. | **6.90** | 6.32–7.54 | **1.26** | 1.12–1.43 | **5.72** | 4.86–6.73 | **1.40** | 1.13–1.73 | **2.83** | 2.32–3.46 | **1.31** | 1.00–1.71 |
| | Only condomless anal int. | **6.88** | 6.50–7.28 | **1.31** | 1.19–1.45 | **5.73** | 5.17–6.35 | **1.47** | 1.24–1.74 | **2.83** | 2.48–3.23 | **1.42** | 1.15–1.77 |
| **Random part** | | $\sigma^2$ | 95%-CI | $\sigma^2$ | 95%-CI | $\sigma^2$ | 95%-CI | $\sigma^2$ | 95%-CI | $\sigma^2$ | 95%-CI | $\sigma^2$ | 95%-CI |
| Survey year | Random Slope | | | 0.10 | 0.06–0.19 | | | 0.01 | 0.00–0.06 | | | 0.16 | 0.08–0.36 |
| 46 countries[1] | Random Intercept | | | 0.32 | 0.20–0.52 | | | 0.31 | 0.18–0.52 | | | 0.04 | 0.01–0.12 |
| Likelihood-ratio test[3] | df, p | | | 25 | p < 0.001 | | | 22 | p < 0.001 | | | 22 | p < 0.001 |
| **Time** | | | | | | | | | | | | | |

(Continued)

**Table 2.** (Continued)

| | | Full sample, excluding only MSM with missing answers | | | | | | | | Sub-sample of MSM screened[2] | | | |
| | | Diagnosis of gonorrhoea/chlamydia, any (N = 262,995) | | | | Diagnosis of gonorrhoea/chlamydia, classified symptomatic (N = 266,407) | | | | Diagnosis of gonorrhoea/chlamydia, classified asymptomatic (N = 47,684) | | | |
| **Fixed part** | | OR | 95%-CI | AOR | 95%-CI | OR | 95%-CI | AOR | 95%-CI | OR | 95%-CI | AOR | 95%-CI |
|---|---|---|---|---|---|---|---|---|---|---|---|---|---|
| Survey year | 2010 | Ref. | | Ref. | 1 | Ref. | | Ref. | 1.00–1.00 | Ref. | | Ref. | |
| | 2017 | **1.79** | 1.74–1.85 | 0.88 | 0.68–1.13 | **1.34** | 1.27–1.42 | 0.83 | 0.60–1.17 | **1.39** | 1.29–1.50 | 0.83 | 0.68–1.01 |
| **Survey artefacts** | | | | | | | | | | | | | |
| French translation | Not used 2010/2017 | Ref. | . | Ref. | | n.a. | . | Ref. | 1.00–1.00 | n.a. | | Ref. | |
| | Used in 2010 | **1.27** | 1.16–1.40 | 0.94 | 0.80–1.10 | 0.91 | 0.76–1.09 | 0.80 | 0.64–1.01 | **2.87** | 2.38–3.45 | **1.53** | 1.14–2.04 |
| | Used in 2017 | | | **1.53** | 1.33–1.75 | | | 0.91 | 0.73–1.13 | | | **3.14** | 2.46–4.01 |
| Discrepant data | No | Ref. | | Ref. | | Ref. | | Ref. | 1.00–1.00 | Ref. | x | Ref. | |
| | Yes | **1.10** | 1.05–1.15 | **1.06** | 1.01–1.12 | 0.99 | 0.91–1.07 | **0.90** | 0.83–0.99 | 0.98 | 0.88–1.09 | 1.02 | 0.91–1.14 |
| **Sample composition** | | | | | | | | | | | | | |
| Age | Per year | 0.99 | 0.99–1.00 | **0.98** | 0.98–0.98 | **0.99** | 0.99–0.99 | **0.98** | 0.97–0.98 | **0.99** | 0.98–0.99 | **0.98** | 0.98–0.98 |
| Settlement size | Village/countryside (<10,000) | Ref. | | Ref. | | Ref. | | Ref. | 1.00–1.00 | Ref. | | Ref. | |
| (inhabitants) | Small town (10,000–99,999) | **1.15** | 1.08–1.23 | **1.14** | 1.06–1.23 | **1.26** | 1.11–1.43 | **1.22** | 1.07–1.39 | 1.01 | 0.87–1.19 | 1.01 | 0.86–1.19 |
| | Medium town (100,000–499,999) | **1.58** | 1.48–1.68 | **1.43** | 1.33–1.53 | **1.66** | 1.47–1.87 | **1.45** | 1.28–1.64 | **1.17** | 1.01–1.35 | 1.13 | 0.97–1.32 |
| | Big city (500,000–999,999) | **2.02** | 1.89–2.16 | **1.64** | 1.53–1.77 | **2.11** | 1.87–2.38 | **1.70** | 1.50–1.93 | **1.18** | 1.01–1.38 | 1.10 | 0.94–1.29 |
| | Very big city (≥1 million) | **2.56** | 2.41–2.73 | **1.94** | 1.81–2.07 | **2.72** | 2.43–3.04 | **2.06** | 1.83–2.31 | **1.26** | 1.10–1.45 | **1.16** | 1.01–1.35 |
| Diagnosed HIV | No | Ref. | | Ref. | | Ref. | | Ref. | 1.00–1.00 | Ref. | | Ref. | |
| | Yes | **3.08** | 2.96–3.20 | **1.89** | 1.80–1.98 | **2.01** | 1.86–2.17 | **1.37** | 1.26–1.49 | 0.92 | 0.84–1.01 | **0.85** | 0.77–0.94 |
| **Testing behaviour** | | | | | | | | | | | | | |
| STI screen | No screening | Ref. | | Ref. | | | | | | | | | |
| | 6–12 months ago | **0.76** | 0.71–0.81 | **0.56** | 0.52–0.60 | | | | | | | | |
| | 1–6 months ago | **1.90** | 1.83–1.97 | 1.03 | 0.99–1.08 | | | | | | | | |
| | Within the last 4 weeks | **3.41** | 3.25–3.57 | **1.55** | 1.47–1.63 | | | | | | | | |
| **Sexual behaviour** | | | | | | | | | | | | | |
| Number of sex partners, | None or one | Ref. | | Ref. | | Ref. | | Ref. | 1.00–1.00 | Ref. | | Ref. | |
| previous 12 months | 2–4 | **1.71** | 1.60–1.83 | **1.97** | 1.78–2.18 | **1.83** | 1.63–2.06 | **1.88** | 1.58–2.24 | **1.33** | 1.14–1.56 | **1.33** | 1.07–1.66 |
| | 5–7 | **2.90** | 2.71–3.11 | **3.21** | 2.87–3.58 | **3.24** | 2.88–3.64 | **3.22** | 2.66–3.88 | **1.53** | 1.30–1.80 | **1.48** | 1.16–1.89 |
| | 8–10 | **4.03** | 3.74–4.34 | **4.21** | 3.75–4.71 | **4.02** | 3.53–4.58 | **3.87** | 3.18–4.71 | **2.04** | 1.72–2.42 | **1.95** | 1.51–2.50 |
| | 11–20 | **5.75** | 5.40–6.12 | **5.91** | 5.31–6.58 | **5.36** | 4.80–5.99 | **5.12** | 4.26–6.16 | **2.32** | 2.00–2.68 | **2.24** | 1.77–2.84 |
| | >20 | **11.16** | 10.51–11.84 | **9.95** | 8.94–11.1 | **9.12** | 8.20–10.13 | **7.99** | 6.65–9.60 | **3.37** | 2.93–3.87 | **3.16** | 2.51–3.99 |
| Anal intercourse, condom use with last non–steady partner | No non–steady partner | Ref. | | Ref. | | Ref. | | Ref. | 1.00–1.00 | Ref. | | Ref. | |
| | No anal intercourse | **2.66** | 2.50–2.83 | **0.70** | 0.63–0.77 | **2.81** | 2.52–3.14 | 0.85 | 0.72–1.02 | **1.69** | 1.46–1.96 | 0.88 | 0.71–1.10 |
| | No condomless anal int. | **3.43** | 3.25–3.63 | **0.85** | 0.77–0.93 | **3.60** | 3.25–3.98 | 1.03 | 0.87–1.22 | **1.71** | 1.50–1.95 | 0.89 | 0.72–1.10 |
| | Some condomless anal int. | **6.90** | 6.32–7.54 | **1.27** | 1.12–1.43 | **5.72** | 4.86–6.73 | **1.40** | 1.13–1.74 | **2.83** | 2.32–3.46 | 1.31 | 1.00–1.71 |
| | Only condomless anal int. | **6.88** | 6.50–7.28 | **1.32** | 1.19–1.45 | **5.73** | 5.17–6.35 | **1.47** | 1.24–1.75 | **2.83** | 2.48–3.23 | **1.40** | 1.13–1.74 |
| **Country-level proportion of survey participants screened in 2010 and 2017** | | | | | | | | | | | | | |
| **In the full samples, screened with a urine-based STI test (or genital swab) and an anal swab in the previous twelve months[4]** | | | | | | | | | | | | | |
| <1.92% | Quartile 1 in 2010 or 2017 | Ref. | | | | Ref. | | | | | | | |
| 1.92–3.25% | Quartile 2 in 2010 | **1.43** | 1.31–1.56 | 1.02 | 0.81–1.29 | 1.12 | 0.96–1.30 | 1.03 | 0.83–1.27 | | | | |
| | Quartile 2 in 2017 | | | 1.18 | 0.92–1.51 | | | 1.32 | 0.88–1.99 | | | | |
| 3.26–7.38% | Quartile 3 in 2010 | **2.15** | 1.95–2.38 | **1.69** | 1.31–2.19 | **1.58** | 1.33–1.87 | 1.29 | 0.99–1.68 | | | | |
| | Quartile 3 in 2017 | | | **1.36** | 1.09–1.69 | | | **1.63** | 1.11–2.40 | | | | |

(Continued)

**Table 2.** (Continued)

| | | Full sample, excluding only MSM with missing answers | | | | | | | | Sub-sample of MSM screened[2] | | | |
| | | Diagnosis of gonorrhoea/chlamydia, any (N = 262,995) | | | | Diagnosis of gonorrhoea/chlamydia, classified symptomatic (N = 266,407) | | | | Diagnosis of gonorrhoea/ chlamydia, classified asymptomatic (N = 47,684) | | | |
| **Fixed part** | | OR | 95%-CI | AOR | 95%-CI | OR | 95%-CI | AOR | 95%-CI | OR | 95%-CI | AOR | 95%-CI |
| 7.39–36.00% | Quartile 4 in 2010 | **2.88** | 2.67–3.09 | **1.70** | 1.28–2.27 | **1.72** | 1.51–1.97 | **1.32** | 1.03–1.69 | | | | |
| | Quartile 4 in 2017 | | | **2.77** | 2.02–3.79 | | | **1.82** | 1.23–2.69 | | | | |
| **Among those screened, reports of anal swabbing as part of the testing intervention** | | | | | | | | | | | | | |
| | No anal swab in 2010 or 2017 | | | | | | | | | Ref. | | | |
| | Anal swab in 2010 | | | | | | | | | **1.61** | 1.49–1.74 | **1.12** | 0.98–1.28 |
| | Anal swab in 2017 | | | | | | | | | | | **1.56** | 1.40–1.73 |
| **Random part** | | σ² | 95%-CI | σ² | 95%-CI | σ² | 95%-CI | σ² | 95%-CI | σ² | 95%-CI | σ² | 95%-CI |
| Survey year | Random Slope | | | 0.10 | 0.05–0.19 | | | 0.00 | 0.00–5.24e +17 | | | 0.13 | 0.06–0.31 |
| 46 countries[1] | Random Intercept | | | 0.09 | 0.05–0.15 | | | 0.19 | 0.10–0.34 | | | 0.03 | 0.01–0.11 |
| Likelihood-ratio test[5] | df, p | | | 31 | p < 0.001 | | | 28 | p = 0.001 | | | 24 | p < 0.001 |

[1] This study includes 46 countries, with four European microstates included in neighbouring (Andorra, Liechtenstein) or surrounding (Monaco, San Marino) countries, and with Albania, Montenegro and Kosovo merged to form a region; this results in 40 country-like entities included in the random part of the model.

[2] Includes only respondents screened with a urine-based STI test (or genital swab) *or* an anal swab.

[3] comparing multilevel model with linear regression model.

[4] grouped by quartiles.

[5] comparing model in Table 2A with model in Table 2B.

increased with larger settlement size (with higher aORs compared to syphilis). Having been diagnosed with HIV was associated with gonorrhoea/chlamydia diagnosis, albeit not as strong as for syphilis. Increased frequency of STI-screening was associated with a higher probability for a gonorrhoea/chlamydia diagnosis (similar as for syphilis). Increasing numbers of sexual partners were stronger associated with higher gonorrhoea/chlamydia diagnosis than seen for syphilis, and declining consistency of condom use was less strongly associated with higher diagnosis probabilities than for syphilis. Similar as for syphilis diagnosis, men who reported no anal intercourse or only condom-protected anal intercourse with the last non-steady partner had a lower risk for gonorrhoea/chlamydia diagnosis than men who had no non-steady partners in the previous twelve months.

Living in a country with screening rates for genital and anal specimens in the third or fourth quartile of country-level screening rates was associated with a higher gonorrhoea/chlamydia diagnosis probability.

**For gonorrhoea/chlamydia classified as symptomatic,** similar to the overall gonorrhoea/chlamydia diagnoses in the previous twelve months model, the MMR model in Table 2A showed a significant difference for the two survey waves after adjustment for all other factors included in the model, while this difference disappeared in the model in Table 2B. Associations with age and settlement size were comparable with the overall gonorrhoea/chlamydia model; there was no impact of the 2017 French language version on symptomatic gonorrhoea/chlamydia. The association of symptomatic gonorrhoea/chlamydia diagnosis with diagnosed HIV was weaker than in the overall model. Associations with partner numbers and condom use with the last non-steady partner were similar for classified symptomatic and overall gonorrhoea/chlamydia diagnosis in the previous twelve months.

Contrasting with the symptomatic syphilis model in Table 1B, where no impact of asymptomatic screening rates on symptomatic diagnosis could be demonstrated, the country-level screening rates for gonorrhoea/chlamydia showed interaction with the survey year and rates were significantly associated with higher aOR for diagnosis of a symptomatic gonorrhoea/chlamydia infection. Living in countries placed in the two highest quartiles of screening rates for asymptomatic gonorrhoea/chlamydia in 2010 and 2017 was associated with increased risks for diagnosis of symptomatic gonorrhoea/chlamydia.

**For gonorrhoea/chlamydia classified as asymptomatic**, the MMR model in Table 2A showed no significant difference for the two survey waves after adjustment for all other factors included in the model, and this remained when including a further screening-related variable in the model in Table 2B, further improving the model. Among the associations with the French language version, discrepant data, age, and settlement size, only the association with the French language version and age, and a weak association with very big cities remained statistically significant. The association with HIV diagnosis usually seen was reversed, suggesting that having been diagnosed with HIV was associated with lower probability of an asymptomatic gonorrhoea/chlamydia diagnosis, than not living with diagnosed HIV. Associations with condom use were similar with the two other gonorrhoea/chlamydia models, and the association with partner numbers was much weaker.

To assess the impact of testing interventions in the model for asymptomatic gonorrhoea/chlamydia the sampling of an anal swab was included in Table 2B as a survey wave interaction term. Having provided an anal swab in 2017 was significantly associated with a higher diagnosis probability for asymptomatic gonorrhoea/chlamydia.

## Discussion

We can demonstrate increases of self-reported overall, symptomatic and asymptomatic infections with syphilis and gonorrhoea/chlamydia in most of the 46 countries covered by EMIS-2010 and EMIS-2017. The interpretation of factors and reasons associated with these increases is challenging because many interacting factors changed: the age structure of the country samples was different, with higher age groups stronger represented in 2017 [31]; the proportion of respondents with diagnosed HIV increased by approximately 35% from 7.7% in 2010 to 10.4% in 2017; STI testing, particularly STI-screening of MSM increased considerably in many countries, both in coverage and frequency [8]; condom use for anal intercourse declined in most countries; and at least among men with multiple non-steady partners, partner numbers increased.

When a statistically significant difference between survey waves is losing significance in a multivariate regression model, this indicates that the explanatory factors included in the model may explain most of the differences observed between the two waves. Notably, the difference between the two survey waves for syphilis is already largely explained by the models not containing the country-level screening rates (Table 1A), suggesting that mainly behaviour change (and to a smaller extent sample composition) is responsible for increasing diagnosis rates. Contrastingly, comparison of Table 2A and 2B suggests that for gonorrhoea/chlamydia testing practices (often but not exclusively reflecting testing policies and guidelines) play a much larger role and appear to contribute more to higher diagnosis rates than behaviour change and sample composition. The role of testing practices would increase even more, if screening frequency, which we considered an individual behaviour in our analysis, would be considered a part of testing interventions. As demonstrated in many publications, routine screening for gonorrhoea and chlamydia in extragenital sites in MSM reveals many, mostly asymptomatic infections. Systematic screening studies suggest that gonorrhoea is almost

equally distributed across genital, rectal and pharyngeal sites, while chlamydia is less frequently identified in the pharynx, and more frequently in the rectum [16]. There is also emerging evidence that most extragenital asymptomatic infections will clear spontaneously within weeks to months [18,32]. Increasing frequency of screening should thus result in more diagnoses. The impact of condoms on transmission of gonorrhoea/chlamydia is highly contested [33], since of all conceivable modes of transmission only the genito-rectal and recto-genital mode would be affected, and our findings would support an only moderate impact of declining condom use on gonorrhoea and chlamydia transmission.

The association of higher partner numbers and declining condom use with higher rates of sexually transmitted infections is an expected and unsurprising finding. For the interpretation of associations with asymptomatic infections, it must be considered that the denominator is different from the denominator for symptomatic infections. The denominator for asymptomatic infections is men who were screened for infections, and thus biased through self-selection to screen for STIs. Considering this, the lower impact of partner numbers on asymptomatic infections is less surprising. One would expect higher screening uptake among men with high partner numbers, because risk perception is generally related rather to the total number of different partners than number of sex acts [34,35].

The associations we found for condom use with the risk of being diagnosed with an STI–for both symptomatic and asymptomatic infections–have been reported elsewhere and are in line with those findings: while condom use provides some level of protection, inconsistent condom use is associated with similar risks as no condom use [36,37]. A stronger association of condom use with symptomatic compared to asymptomatic infections is likely explained by a higher probability of genital infections to cause recognizable symptoms that could be prevented by condom use.

Since syphilis is more frequently found in men diagnosed with HIV than gonorrhoea/chlamydia, the stronger association of syphilis compared to gonorrhoea/chlamydia with diagnosed HIV had to be expected and appears in line with published observations for respondents with a last negative *vs.* positive HIV test result [28,29]. The fact that an opposite association can be observed for asymptomatic gonorrhoea/chlamydia is surprising at first. Again, this may be related to self-selection for high STI risk among respondents presenting for STI screening. While people with diagnosed HIV may be screened as part of regular HIV care irrespective of sexual risks, respondents without diagnosed HIV who undergo frequent STI screening may be a group with high risk for STIs. It has been reported that e.g. men with a negative HIV test result for HIV who take HIV pre-exposure prophylaxis have an equally high [38,39] or even higher [8] probability to be screened and diagnosed with asymptomatic STIs as men living with HIV.

The different impact of settlement size on syphilis and gonorrhoea/chlamydia diagnoses is likely due to the different prevalence of the infections: sexual networks may be more important than settlement size for syphilis due to the higher prevalence of syphilis in specific subgroups, while gonorrhoea and chlamydia infection are more widespread in the general MSM population [40].

A higher impact of the French translation issue on the odds for diagnosis of asymptomatic infections is highly plausible. If respondents understood the question as whether they had been tested for and not diagnosed with the respective infection, the probability to over report infections would be higher when testing was not triggered by symptoms. We would like to emphasize that we controlled for a potential overestimation of STI diagnoses in French questionnaires.

Finally, further associations remain to be explained: age and the country-level variables, particularly how the rates of diagnosed asymptomatic infections with gonorrhoea/chlamydia

might impact symptomatic infections with gonorrhoea/chlamydia, and the different effect of anal swabs on diagnosis of asymptomatic gonorrhoea/chlamydia in 2010 and 2017.

We believe that the growing effect of anal swabbing on the diagnosis of asymptomatic gonorrhoea/chlamydia 2017 as compared to 2010 reflects the increased sampling of pharyngeal swabs for gonorrhoea/chlamydia in addition to anal swabbing since 2010. While the sampling of pharyngeal swabs had not been queried in the questionnaire, sampling of pharyngeal swabs is almost always accompanied by sampling of anal swabs (but not vice versa). The inclusion of the anal swab survey wave interaction term might therefore indicate the effect of additional increased pharyngeal swabbing on the diagnosis of asymptomatic gonorrhoea (and to a much lesser extent chlamydia).

For gonorrhoea and chlamydia, two infections that are often asymptomatic and clear spontaneously by not well-understood immune responses [17,41–43], the association between symptomatic infections and country-level rates of screening for asymptomatic infections with gonorrhoea/chlamydia would be compatible with the arrested immunity hypothesis [44,45]. According to this hypothesis, early diagnosis and treatment of asymptomatic infections might abrogate the development of an effective immune response to the pathogen and thus may prevent the development of a status of decreased susceptibility to re-infection and subsequent susceptibility to re-infection. We identified this association between increased asymptomatic screening and increased symptomatic infections in the model for symptomatic gonorrhoea/chlamydia. Declining probability for infection with increasing age in this scenario might also reflect the development of a status of decreased susceptibility through repeated lifetime exposures, asymptomatic infections and subsequent clearance.

An alternative explanation for the association with country-level screening could be residual confounding/collinearity between the country-level variables and high rates of symptomatic gonorrhoea/chlamydia. However, we believe that the lack of association of screening rates for syphilis on symptomatic syphilis does not support this explanation.

We cannot claim that our data provide conclusive evidence that high rates of asymptomatic screening contribute to increasing rates of symptomatic gonorrhoea and chlamydia infection, because our data cannot prove causality, and we cannot exclude residual confounding. At the same time, we can affirm that our data do not provide evidence that increased screening for asymptomatic gonorrhoea and chlamydia infections reduces the prevalence of symptomatic infections with these two bacteria–although this is the public health gain that is promised for increased screening. One could only argue that the prevalence of gonorrhoea and chlamydia might be even higher without screening and that we have not screened enough to see the desired effects. With close to complete and highly frequent screening of the MSM population for asymptomatic gonorrhoea and chlamydia it might theoretically be possible to eliminate these two infections among MSM–but it is highly doubtful whether such levels of large-scale high frequency screening would ever be feasible and sustainable [46].

Even if our data are inconclusive, they raise serious doubts about the promises of increased screening for gonorrhoea and chlamydia. If there is some truth in the arrested immunity hypothesis, it is conceivable that increased screening for gonorrhoea/chlamydia may do more harm than good, both on the individual and the public health level–and our data would be compatible with this view [47]. So far there is a noticeable lack of published evidence for individual or public health benefits of increased gonorrhoea/chlamydia screening among MSM. It is time to critically review the evidence base and assess the current move to further increase screening. More research and less assumption-based approaches are necessary.

The conclusions regarding syphilis are different. For all reported syphilis infections, we observed a significant association with increasing screening rates, but we saw no effect of screening rates on rates of symptomatic syphilis. Contrary to gonorrhoea and chlamydia,

serious long-term sequelae after varying periods without clinical symptoms represent the typical course of syphilis. Testing for asymptomatic infections is therefore advised to prevent possible long-term sequelae. Even if delaying testing and treatment may induce a stronger immune response against syphilis and thus may attenuate symptoms when re-infections occur [48], the risk of progressive disease is likely to be too high to dispense regular testing and treatment of active infections.

## Limitations

There are several limitations to our analysis. EMIS-2010 and EMIS-2017 were both large low-threshold cross-sectional online surveys: All diagnosis and behaviour data are self-reported and thus can be subject to social desirability bias, recall bias, erroneous attribution of symptoms to an infection diagnosis, and confusion of different STIs. The question regarding symptoms related to the last STI test and not specifically to the diagnosed STIs. We cannot rule out that symptoms were caused by STIs not queried (such as *e.g. M.genitalium*) or omitted from this analysis (such as *e.g.* anal/genital warts). When people had more than one STI test in the previous twelve months, infections diagnosed during these other tests could not be classified as symptomatic or asymptomatic. Also, the answer to this question could be missing or the question could have been misunderstood or misinterpreted. As a result, between 50% and 55% of the reported infections in the two survey rounds could not be classified either as symptomatic or asymptomatic. A response to STI screening was missing from 5.5% of the respondents in 2010 and from 3.6% in 2017, and across demographic and behavioural variables non-response was associated with factors that are also associated with a lower probability for being tested and diagnosed with an STI. The questionnaires did not query whether pharyngeal swabs were applied, thus we can only indirectly determine how much pharyngeal swabbing has increased and contributed to (mostly asymptomatic) diagnoses of gonorrhoea and chlamydia. We cannot determine the location of these infections, and we do not know whether the symptoms reported by the respondents were related to the infections that have been diagnosed. For syphilis, we do not know which proportion of the diagnoses that were reported were indeed active infections requiring treatment, because we have not queried STI treatment. We also do not know which tests have been used to diagnose the infections, and whether there are relevant differences regarding sensitivity and specificity of tests across countries. For example, some respondents may report a reactive rapid test for syphilis antibodies as a syphilis diagnosis, while others only report a confirmed active infection. It is conceivable that the increasing use of PrEP among MSM has an impact on STI diagnoses that is not fully explained by partner numbers and screening frequency. While PrEP use was assessed in 2017, there were no questions on PrEP in 2010, and the PrEP variable was thus not part of our dataset. The potential impact of PrEP use on syphilis diagnoses will be analysed in a separate publication that is currently in preparation.

## Conclusions

We observed a growing number of bacterial STI diagnoses among respondents of two large online surveys for MSM between 2010 and 2017 in most, but not all European countries. Distinguishing between symptomatic infections and asymptomatic infections demonstrates that increased uptake, frequency, and comprehensiveness of screening contributed considerably to these increases. In addition, behaviour changes such as declining condom use, and increasing numbers of sexual partners, contributed to increasing STI transmission.

Our data suggest that increasing screening frequency and the proportion of MSM screened for gonorrhoea/chlamydia may not help to reduce the numbers of symptomatic infections; it

may even contribute to a further increase. A conceivable biological explanation for such a paradoxical effect could be the disruption of immunological clearance of asymptomatic infections with a subsequent state of reduced susceptibility to re-infection by early detection and treatment and a rapid re-establishment of susceptibility to re-infections.

More research on the natural history of asymptomatic gonorrhoea/chlamydia cases and on beneficial and adverse effects of this STI control strategy is urgently needed before adopting a "search-and-treat" strategy for these two infections among MSM. Abandoning screening and treatment would of course raise new questions: how should sexual partners of men with symptomatic gonorrhoea/chlamydia infection be managed? Many PrEP guidelines use the diagnosis of STIs, particularly rectal STIs, as an indication to discuss HIV-PrEP. The overarching question remains what the consequences of not diagnosing many of these STIs are (from both individual and public health perspectives).

## Supporting information

**S1 Table. Syphilis, self-reported diagnoses in the previous 12 months.**
(DOCX)

**S2 Table. Gonorrhoea/Chlamydia (Gon/Chl), self-reported diagnoses in the previous 12 months.**
(DOCX)

**S3 Table. Partner numbers, condom use with the last non-steady partner, and STI screen in the previous six months.**
(DOCX)

## Acknowledgments

We thank all study participants and collaborators for being part of something huge. EMIS-2017 is coordinated by Sigma Research at the London School of Hygiene and Tropical Medicine (LSHTM) in association with the Robert Koch Institute (RKI) in Berlin. The following list acknowledges all partners in EMIS by country. Individual names are mentioned if a freelancer was the main contact and/or translator or where input on the questionnaire development came from a person not formally representing an organisation. The order (if available) is main NGO partner, other NGO partners, academic partners, governmental partners, individuals. **Europe**: PlanetRomeo, European AIDS Treatment Group (EATG), Eurasian Coalition on Male Health (ECOM), European Centre for Disease Prevention and Control (ECDC), European Monitoring Centre for Drugs & Drug Addiction (EMCDDA), European Commission (DG SANTE). **AL**: Arian Boci. **AT**: Aids Hilfe Wien, Dr Frank M. Amort. **BA**: lgbti.ba, Masha Durkalić. **BE**: SENSOA, exaequo, Observatoire du SIDA et des sexualités, Sciensano. **BG**: HUGE, GLAS Foundation, Dr Emilia Naseva, Petar Tsintsarski. **BY**: Vstrecha. **CA**: Health Initiative for Men, Rézo, Gay Men's Sexual Health Alliance of Ontario, CATIE, Ontario HIV Treatment Network, Université du Quebec & Montréal, University of Toronto, Ryerson University, University of Windsor, University of Victoria, Public Health Agency of Canada, Rob Gair. **CH**: Swiss AIDS Federation, Cantonal Hospital St. Gallen, Centre Hospitalier Universitaire Vaudois, University Hospital Zurich, Swiss Federal Office of Public Health. **CY**: AIDS Solidarity Movement. **CZ**: AIDS pomoc, National Institute of Public Health, Tereza Zvolska, Dr Michal Pitoňák. **DE**: Deutsche AIDS-Hilfe, Robert Koch Institute, BZgA, Dr Michael Bochow, Dr Richard Lemke. **DK**: AIDS-Fondet, Statens Serum Institut, François Pinchon, Jakob Haff. **EE**: Eesti LGBT, VEK LGBT, Estonia National Institute for Health Development, Dr Kristi Rüütel. **ES**: Stop Sida, CEEISCAT, Ministerio de Sanidad. **FI**: Positiiviset, Hivpoint,

SeksiPertti, Trasek, National Institute for Health and Welfare. **FR**: AIDES, Coalition PLUS, SexoSafe, Santé Publique France, INSERM. **GR**: Ath Checkpoint, Thess Checkpoint; Positive Voice. **HR**: Iskorak, gay.hr, Zoran Dominković, Vjeko Vacek. **HU**: Háttér Society, Tamás Bereczky. **IE**: Gay Health Network, Man2Man, HIV Ireland, Outhouse, GOSHH, Sexual Health Centre Cork, AIDSWEST, Gay Community News, Health Service Executive, Gay Men's Health Service, Sexual Health and Crisis Pregnancy Programme, Health Protection Surveillance Centre. **IL**: Israel AIDS Task Force, Israel Ministry of Health, Dr Zohar Mor. **IS**: Samtökin'78. **IT**: Arcigay, Fondazione LILA Milano ONLUS, University of Verona, Dr Raffaele Lelleri. **LB**: SIDC, Dr Ismaël Maatouk. **LT**: demetra, LGL, Gayline. **LV**: Testpunkts, Baltic HIV Association, Dr Antons Mozalevskis, Indra Linina. **MD**: GENDERDOC-M. **ME**: Juventas. **MK**: Subversive Front, Dr Kristefer Stojanovski. **MT**: Malta LGBTIQ Rights Movement, Allied Rainbow Communities, Infectious Disease Prevention and Control Unit, Silvan Agius, Russel Sammut. **NL**: Results in Health, Maastricht University, Amsterdam Pink Panel, Soa Aids Nederland, Rutgers, Dr Wim Vanden Berghe, Marije Veenstra. **NO**: Helseutvalget, Norwegian Directorate of Health, Folkehelseinstituttet, Dr Rigmor C. Berg. **PH**: Bisdak Pride-Cebu, Cebu Plus, HASH, Pinoy Plus, UP Babaylanes, YPEER, TLF, Office of the WHO Representative in the Philippines, Natasha Montevirgen, Mikael N. Navarro. **PL**: Spoleczny Komitet ds AIDS, Kampania Przeciw Homofobii, Lambda Warszawa, Dr Łukasz Henszel. **PT**: GAT Portugal, CheckpointLX, Associação ABRAÇO, rede ex aequo, SexED, dezanove, ILGA Portugal, Trombeta Bath, ISPUP. **RO**: Association "Eu sunt! Tu?", PSI Romania, ARAS Romania, Tudor Kovacs. **RS**: Association Duga, Association Red Line, Omladina JAZAS-a Novi Sad, Institute of Public Health of Serbia, Sladjana Baros, Dr Marija Pantelic. **RU**: The Charity Foundation For Support of Social Initiatives and Public Health/LaSky Project. **SE**: RFSL, University of Gothenburg, Folkhålsomyndigheten. **SI**: ŠKUC, Legebitra, LJUDMILA. **SK**: PRIDE Košice, Light-House Slovakia, Slovak Medical University, Public Health Authority of the Slovak Republic, Dr Zuzana Klocháňová. **TR**: Pozitif Yaşam, Sami S. Yazıcılaroğlu. **UA**: Alliance for Public Health, alliance.global, msmua.org, Oleksii Shestakovskyi. **UK**: Terrence Higgins Trust, NAM, PrEPster, Antidote, Horizon Drugs and Alcohol Support, LGBT Foundation, Yorkshire MESMAC, MESMAC Newcastle, Derbyshire LGBT+, Trade Sexual Health, London Friend, GMFA, Spectra, International HIV Partnerships, International Planned Parenthood Federation, Bristol University, University College London, Sigma Research, Raul Soriano. **Other**: Dr John Pachankis, Dr Mark Hatzenbühler, Dr Valeria Stuardo Ávila, Dr Michael W. Ross.

## Author Contributions

**Conceptualization:** Ulrich Marcus.

**Data curation:** Axel J. Schmidt.

**Formal analysis:** Ulrich Marcus, Massimo Mirandola.

**Funding acquisition:** Ulrich Marcus.

**Investigation:** Ulrich Marcus, Axel J. Schmidt.

**Methodology:** Ulrich Marcus, Massimo Mirandola, Axel J. Schmidt.

**Project administration:** Susanne B. Schink, Axel J. Schmidt.

**Supervision:** Axel J. Schmidt.

**Validation:** Massimo Mirandola, Lorenzo Gios.

**Visualization:** Susanne B. Schink.

**Writing – original draft:** Ulrich Marcus, Axel J. Schmidt.

**Writing – review & editing:** Massimo Mirandola, Susanne B. Schink, Lorenzo Gios, Axel J. Schmidt.

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
