## [Decision Letter · Decision Letter 0]

23 Dec 2020

PONE-D-20-35441

Changes in the prevalence of self-reported sexually transmitted bacterial infections from 2010 and 2017 in two large European samples of men having sex with men – Is it time to re-evaluate STI-screening as a control strategy?

PLOS ONE

Dear Dr. Marcus,

Thank you for submitting your manuscript to PLOS ONE. After careful consideration, we feel that it has merit but does not fully meet PLOS ONE’s publication criteria as it currently stands. Therefore, we invite you to submit a revised version of the manuscript that addresses the points raised during the review process.

Apologies that I was unable to secure a second reviewer in a timely fashion. However, from my own reading of the manuscript I agree with the comments made by the single reviewer. Please consider each carefully and incorporate as appropriate.

We look forward to receiving your revised manuscript.

Kind regards,

Ethan Morgan

Academic Editor

PLOS ONE

Journal Requirements:

2. You indicated have not indicated whether ethical approval was obtained or waived for the EMIS-2010 survey. We understand that the framework for ethical oversight requirements for studies of this type may differ depending on the setting and we would appreciate some further clarification regarding your research. Could you please provide further details on whether approval or a waiver was granted for the EMIS-2010 survey data  collection. Please include  this information in the manuscript and submission form under "Ethics Statement".

3.We note that you have indicated that data from this study are available upon request. PLOS only allows data to be available upon request if there are legal or ethical restrictions on sharing data publicly. For information on unacceptable data access restrictions, please see http://journals.plos.org/plosone/s/data-availability#loc-unacceptable-data-access-restrictions.

Reviewers' comments:

Reviewer's Responses to Questions

**Comments to the Author**

1. Is the manuscript technically sound, and do the data support the conclusions?

Reviewer #1: Partly

2. Has the statistical analysis been performed appropriately and rigorously? 

Reviewer #1: Yes

3. Have the authors made all data underlying the findings in their manuscript fully available?

Reviewer #1: No

4. Is the manuscript presented in an intelligible fashion and written in standard English?

Reviewer #1: Yes

5. Review Comments to the Author

Reviewer #1: This is an interesting manuscript and I enjoyed reading the results. There are some minor comments regarding introduction and methods and some major concerns regarding the discussion. Overall, I believe that the authors are able and capable of addressing these concerns.

Introduction

The introduction is very short but describes the current body of evidence well and is highly relevant for most questions tackled in the manuscript at hand. However, at times, this part of the manuscript appears disjointed with a lack of transition between individual paragraphs – especially the ‘observations’ at the end of the introduction are at a little surprising as they appear to be results with some of these not being addressed sufficiently in the introduction itself (e.g., the discussion on condomless sex is very limited in the discussion). Further to this point, it would be helpful to provide more detail here given the sample includes a large number of countries with potentially different epidemiological profiles regarding the use of condoms.

Methods

The methods of the survey itself are well-explained and the reference to elsewhere is appropriate. The summary provides sufficient information for readers to understand the methodology of the survey itself.

I agree with the grouping of variables authors have undertaken for primary outcomes.

I recommend to reword the ‘Secondary outcomes’ section; this is currently cumbersome to read.

I am confused by the usage of the term ‘policy’ as the manuscript does not analyse testing policies and procedures in different countries but relies on self-reported data. This data may represent local cultures in STI testing and – depending on the analysis and interpretation – also concerning the performance of test, but it unlikely to be able to capture STI testing policies.

The ‘hierarchical list of variables’ appears to refer to Hierarchical Segmentation Analysis? If this is the case, please provide further information on the use of this method in context to the data set at hand.

NOTE: Some of the figures included were difficult to interpret due to low quality in the file provided by PloSOne. It is unclear if this happened during the submission process or if the quality was limited from the beginning.

Results

I note that missing responses were discussed in the first paragraph of the results section. Was a missing data analysis conducted to see if there are any differences on main demographic variables between these groups?

The result section is clear and presents a thorough (and interesting) analysis.

Discussion

I agree with the authors that the interpretation of trends and changes is challenging. I was wondering if the inclusion of a case study as part of the results/discussion might be helpful to foster a better understanding of this complexity. E.g., by describing these changes for one country in detail. It might generally be a good idea to provide more in-depth information regarding some of the rather general statements made in the discussion (e.g., ‘STI testing, particularly STI-screening of MSM increased considerable in many countries…”). At this stage large parts of the discussion appear to be an extension of the result section or a further data exploration rather than a critical discussion and contextualisation of the findings. Furthermore, there are concepts briefly mentioned in the discussion that are not contextualised sufficiently and at times with a limited critical perspective (e.g., some of the assumptions associated with risk perception and subsequent behaviour).

I again note the usage of the term ‘policy’ in the discussion and question it’s usage.

6. PLOS authors have the option to publish the peer review history of their article (what does this mean?). If published, this will include your full peer review and any attached files.

Reviewer #1: No

---

## [Author Response · Author response to Decision Letter 0]

14 Jan 2021

Response to the Editors 

Journal Requirements:

Response: We have re-formatted the manuscript to meet the PLOS ONE style requirements. 

2. You have not indicated whether ethical approval was obtained or waived for the EMIS-2010 survey. We understand that the framework for ethical oversight requirements for studies of this type may differ depending on the setting and we would appreciate some further clarification regarding your research. Could you please provide further details on whether approval or a waiver was granted for the EMIS-2010 survey data collection. Please include this information in the manuscript and submission form under "Ethics Statement".

Response: We apologise for this oversight. The respective section now reads: 

Ethics approval was granted by the Research Ethics Committee of the University of Portsmouth for EMIS-2010 (REC application number 08/09:21), and by the Ethics Committee of the London School of Hygiene and Tropical Medicine for EMIS-2017 (reference 14421/RR/8805).

3.We note that you have indicated that data from this study are available upon request. PLOS only allows data to be available upon request if there are legal or ethical restrictions on sharing data publicly. For information on unacceptable data access restrictions, please see http://journals.plos.org/plosone/s/data-availability#loc-unacceptable-data-access-restrictions.

Response: As indicated in the submission form, 

The EMIS-2017 dataset used for this analysis has been obtained from the London School of Hygiene and Tropical Medicine under a data transfer agreement that prohibits to sharing the dataset publicly. Although we cannot make study data publicly accessible at the time of publication, all authors commit to make the data underlying the findings of the study available in compliance with the PLOS Data Availability Policy. Data requests should be addressed to the London School of Hygiene and Tropical Medicine Research Operations Office Data Management Lead: alex.hollander@lshtm.ac.uk, the first author (MarcusU@rki.de), and the Principal Investigator of EMIS-2017 (Peter.Weatherburn@lshtm.ac.uk). Individuals requesting data should present their research objective(s) and enclose a list of requested variables. To protect the confidentiality of participants, data sharing is contingent upon appropriate data handling and good scientific practice by the person requesting the data and should furthermore be in accordance with all applicable local requirements. The London School of Hygiene and Tropical Medicine administrative offices are located at Keppel Street, London WC1E 7HT, United Kingdom.

Response to the Reviewer

Reviewer #1: This is an interesting manuscript and I enjoyed reading the results. There are some minor comments regarding introduction and methods and some major concerns regarding the discussion. Overall, I believe that the authors are able and capable of addressing these concerns.

Response: We thank the reviewer for their positive evaluation and comments.

Introduction

The introduction is very short but describes the current body of evidence well and is highly relevant for most questions tackled in the manuscript at hand. However, at times, this part of the manuscript appears disjointed with a lack of transition between individual paragraphs – especially the ‘observations’ at the end of the introduction are at a little surprising as they appear to be results with some of these not being addressed sufficiently in the introduction itself (e.g., the discussion on condomless sex is very limited in the discussion). Further to this point, it would be helpful to provide more detail here given the sample includes a large number of countries with potentially different epidemiological profiles regarding the use of condoms.

Response: As suggested, we addressed the issues in the Introduction and re-phrased the last paragraph.

Methods

The methods of the survey itself are well-explained and the reference to elsewhere is appropriate. The summary provides sufficient information for readers to understand the methodology of the survey itself.

I agree with the grouping of variables authors have undertaken for primary outcomes.

I recommend to reword the ‘Secondary outcomes’ section; this is currently cumbersome to read.

Response: We have reworded the ‘Secondary outcomes’ section to improve the flow of the argument.

I am confused by the usage of the term ‘policy’ as the manuscript does not analyze testing policies and procedures in different countries but relies on self-reported data. This data may represent local cultures in STI testing and – depending on the analysis and interpretation – also concerning the performance of test, but it unlikely to be able to capture STI testing policies.

Response: We thank the reviewer for this comment. We have now introduced a careful distinction between the terms ‘policy’ and ‘practices’.

The ‘hierarchical list of variables’ appears to refer to Hierarchical Segmentation Analysis? If this is the case, please provide further information on the use of this method in context to the data set at hand.

Response: The use of the term ‘hierarchical’ may be misleading in this context and we deleted it. The list of variables was chosen on theoretical assumptions, and we arranged them in the regression analysis in the following order: 1) Time (survey wave); 2) Survey artefacts (language, discrepant data); 3) Sample composition/demographics (age, settlement size, HIV diagnosis); 4) Testing behavior (screening recency used as surrogate for screening frequency); 5) Sexual behavior (partner number, type of partner, anal intercourse with the last non-steady partner, and condom use with the last non-steady partner); 6) country-level intervention practices (proportion of respondents screened). No Hierarchical Segmentation Analysis was used.

NOTE: Some of the figures included were difficult to interpret due to low quality in the file provided by PloSOne. It is unclear if this happened during the submission process or if the quality was limited from the beginning.

Response: We used the tools provided by PLOS ONE to create the figures, and the uploaded figures were of high quality. We do not know what happens during the submission process and would appreciate input from the editors or technical team to make sure that the quality of the figures is maintained.

Results

I note that missing responses were discussed in the first paragraph of the results section. Was a missing data analysis conducted to see if there are any differences on main demographic variables between these groups?

Response: We have added a short description of a missing data analysis in the limitations section. In summary, respondents with missing responses on screening are likely to have a lower probability for having received an STI diagnosis than respondents who answered these questions.

The result section is clear and presents a thorough (and interesting) analysis.

Discussion

I agree with the authors that the interpretation of trends and changes is challenging. I was wondering if the inclusion of a case study as part of the results/discussion might be helpful to foster a better understanding of this complexity. E.g., by describing these changes for one country in detail. It might generally be a good idea to provide more in-depth information regarding some of the rather general statements made in the discussion (e.g., ‘STI testing, particularly STI-screening of MSM increased considerable in many countries…”). At this stage large parts of the discussion appear to be an extension of the result section or a further data exploration rather than a critical discussion and contextualisation of the findings. Furthermore, there are concepts briefly mentioned in the discussion that are not contextualised sufficiently and at times with a limited critical perspective (e.g., some of the assumptions associated with risk perception and subsequent behaviour).

I again note the usage of the term ‘policy’ in the discussion and question its usage.

Response: As mentioned above we have replaced the term ‘policy’ by ‘practices’ where appropriate. We have prepared a case study, describing in detail the changes between the two survey waves for a single country. This case study would add a minimum of 500 words to the manuscript. We are not convinced that including a case study in the manuscript would be useful. Various factors changed in numerous countries to different extents. Details about changes between the two survey waves for all countries are contained in the figures and supplementary tables. An appropriate way to deal with this complexity is the use of multivariate multilevel regression analyses, in which we control e.g. for the effects of changes in the sample composition and survey artefacts such as the French translation issue, as we presented. We hope that we can resolve the issue of the quality of the uploaded figures through input from the technical team at PLOS ONE, so that graphics will be of optimal quality/resolution/file size. We have extended contextualization and the discussion of some of the findings where applicable. Thank you.

---

## [Decision Letter · Decision Letter 1]

2 Mar 2021

Changes in the prevalence of self-reported sexually transmitted bacterial infections from 2010 and 2017 in two large European samples of men having sex with men – Is it time to re-evaluate STI-screening as a control strategy?

PONE-D-20-35441R1

Dear Dr. Marcus,

We’re pleased to inform you that your manuscript has been judged scientifically suitable for publication and will be formally accepted for publication once it meets all outstanding technical requirements.

Kind regards,

Zixin Wang, PhD.

Academic Editor

PLOS ONE

Additional Editor Comments (optional):

Reviewers' comments:

Reviewer's Responses to Questions

**Comments to the Author**

1. If the authors have adequately addressed your comments raised in a previous round of review and you feel that this manuscript is now acceptable for publication, you may indicate that here to bypass the “Comments to the Author” section, enter your conflict of interest statement in the “Confidential to Editor” section, and submit your "Accept" recommendation.

Reviewer #1: All comments have been addressed

2. Is the manuscript technically sound, and do the data support the conclusions?

Reviewer #1: Yes

3. Has the statistical analysis been performed appropriately and rigorously? 

Reviewer #1: Yes

4. Have the authors made all data underlying the findings in their manuscript fully available?

Reviewer #1: No

5. Is the manuscript presented in an intelligible fashion and written in standard English?

Reviewer #1: Yes

6. Review Comments to the Author

Reviewer #1: The authors have done a good job of revising the manuscript in line with my suggestions. I have no further comments.

7. PLOS authors have the option to publish the peer review history of their article (what does this mean?). If published, this will include your full peer review and any attached files.

Reviewer #1: No

---

## [Editor Report · Acceptance letter]

5 Mar 2021

PONE-D-20-35441R1 

Changes in the prevalence of self-reported sexually transmitted bacterial infections from 2010 and 2017 in two large European samples of men having sex with men – is it time to re-evaluate STI-screening as a control strategy? 

Dear Dr. Marcus:

I'm pleased to inform you that your manuscript has been deemed suitable for publication in PLOS ONE. Congratulations! Your manuscript is now with our production department. 

Kind regards, 

on behalf of

Professor Zixin Wang 

Academic Editor

PLOS ONE